# Human cerebrospinal fluid net flow enhanced by respiration during the awake state

Cerebrospinal fluid dynamics play a crucial role in maintaining brain homeostasis by delivering nutrients, transmitting immune signals, and clearing waste products. While cardiac activity primarily drives the pulsatile movement of cerebrospinal fluid, respiration has been shown to facilitate low-frequency oscillations and contribute to bulk flow. Recent studies suggest that enhancing respiratory function may be an effective intervention to modulate cerebrospinal fluid dynamics. This study included 20 individuals with long-term formal training in Seokmun Hoheup, a lower belly–centered breathing practice (mean $\pm$ SD age, 58.1 $\pm$ 17.3 years; 8 females), and 25 controls with no formal long-term breathing practice (mean $\pm$ SD age, 49.2 $\pm$ 20.2 years; 12 females). All underwent real-time velocity-encoding magnetic resonance imaging to assess cerebrospinal fluid movement at the foramen magnum and lateral ventricle during both regular breathing and deep breathing. Deep breathing enhances cerebrospinal fluid dynamics in both groups, increasing displacement and net flow, particularly at the foramen magnum. Seokmun Hoheup trained participants show greater cerebrospinal fluid movement than controls at both the foramen magnum and lateral ventricle. Even during regular breathing, trained participants show higher cerebrospinal fluid mean speed, displacement, and net flow. Among respiratory factors, inhale length and diaphragm displacement show the strongest correlations with cerebrospinal fluid movement. Respiration modulated cerebrospinal fluid dynamics through both mechanical enhancement of venous outflow and autonomic modulation of the heart, with mechanical effects predominating in the lateral ventricle and both pathways contributing to the foramen magnum. Our findings identify respiration in the awake state as a modifiable, noninvasive mechanism that influences involuntary functions such as cerebrospinal fluid dynamics and may have implications for cerebrospinal fluid-mediated brain homeostasis.

Cerebrospinal fluid (CSF) maintains brain homeostasis by distributing nutrients, hormones, and immune cells, as well as clearing waste from interstitial spaces[1-4]. Pulsatile flow driven by cardiac activity is considered the primary factor influencing CSF dynamics[5-7]. Specifically, CSF velocity peaks closely align with arterial wall movements, thereby modulating fluid exchange in interstitial spaces[7]. Respiration plays a significant role in CSF dynamics, facilitating low-frequency oscillations and bulk flow[8-12]. These respiratory effects are thought to be mediated

✉e-mail: cogswell.petrice@mayo.edu; min.paul@mayo.edu

in part by venous return, which has been proposed as a key driver of pressure and flow modulation within the CSF system[9,13,14].

Recent research suggests that respiratory function may be a viable intervention to modulate CSF flow[11,15]. Unlike cardiac activity, which is purely involuntary, breathing combines both voluntary and involuntary control. This dual nature makes it difficult to study in animal models, leaving the underlying respiratory mechanisms in CSF dynamics relatively underexplored and highlighting the need for human studies. Advances in velocity-encoding magnetic resonance imaging (MRI) techniques with high temporal and spatial resolution now allow direct measurement of both cardiac- and respiratory-driven components of CSF flow[12], providing a precise method to assess respiration as a potential modulator[16,17].

While most human studies have focused on oscillatory CSF movements[9,11,12], it remains unclear how these oscillations translate into net flow[18–20]. This gap is particularly important given that CSF dynamics are recognized as a major contributor to glymphatic flow, the system responsible for waste clearance in the brain[7,21]. Human studies have reported respiration-driven CSF net flow, but insight into the detailed mechanisms underlying this phenomenon remains limited[9,22]. Nevertheless, the increasing popularity of well-being practices that incorporate breathing provides an opportunity to explore the effects of long-term respiratory training on CSF dynamics[23–25].

In this study, we first validated the velocity-encoding MRI technique using a flow phantom to ensure measurement accuracy and reproducibility. We then examined two groups: individuals trained in Seokmun Hoheup, a lower belly–centered breathing practice (trained group, T)[26], and individuals with no prior long-term structured breathing training (non-trained group, NT). All participants underwent CSF flow measurement at the foramen magnum (FM), located at the craniospinal junction, and in the lateral ventricle (LV) using velocity-encoding MRI during both regular breathing (RB) and voluntary slow, deep breathing (DB) acquired during wakefulness. To further understand how respiration shapes CSF dynamics, we modeled individual diaphragm motion and identified key respiratory features associated with CSF flux. This study investigates how respiration-induced physiological changes modulate CSF movements and evaluates the potential of respiration as a modifiable and trainable mechanism.

## Results

### Baseline demographics and physiological characteristics

The study included 20 T participants and 25 NT participants, as summarized in the experimental flow chart (Fig. S1). Group demographics are presented in Table S1. Age did not differ significantly between groups (T: 58 ± 17.3 years; NT: 49 ± 20.2 years; $p = 0.128$, Table S1). Sex distribution was comparable (40% female in T vs. 48% in NT, $p = 0.764$). Cardiovascular measures, including systolic ($p = 0.457$) and diastolic blood pressure (BP) ($p = 0.140$), as well as heart rate (HR) during RB ($p = 0.637$) and DB ($p = 0.699$), showed no significant differences between the T and NT groups. Within-subject comparisons of HR between breathing conditions revealed a small but significant increase in the T group from RB (64.2 ± 8.04 bpm) to DB (66.0 ± 9.16 bpm, $p = 0.006$), whereas no significant change was observed in the NT group from RB (65.4 ± 8.91 bpm) to DB (67.1 ± 9.07 bpm, $p = 0.094$). Respiratory rate (RR) was significantly lower in the T group compared with the NT group during RB (T-RB: 8.4 ± 3.26 vs. NT-RB: 14.1 ± 3.26 breaths/min, $p = 0.013$) and DB (T-DB: 4.8 ± 2.33 vs. NT-DB: 6.9 ± 3.06 breaths/min, $p = 0.014$). Within-group comparisons confirmed that RR decreased significantly during DB compared with RB in both groups ($p < 0.0001$).

### Validation of net flow measurements using a precision flow phantom

To verify the accuracy and reliability of phase-contrast MRI (PC-MRI) quantification, a flow phantom experiment was performed (Fig. S2,

3). PC-MRI successfully captured both cardiac-like and respiratory-like oscillatory flow conditions, with mean speed, displacement, and net flow closely matching the imposed inputs (Fig. S2). Under cardiac-like oscillations, increasing flow input amplitude led to higher mean speed and net flow, whereas displacement did not show a consistent dependence on flow input amplitude. In contrast, respiratory-like oscillations produced significant increases in mean speed and displacement with higher flow amplitudes (Kruskal–Wallis test, H = 7.2, $p = 0.0036$), with the largest difference detected between condition 1 (1.5 cm/s, 8-s cycle) and condition 3 (2.5 cm/s, 8-s cycle; post hoc $p = 0.0219$). Net flow also showed a significant overall effect across respiratory-like conditions (Kruskal–Wallis test, H = 5.69, $p = 0.029$), with condition 1 versus 3 approaching significance ($p = 0.051$), indicating a trend toward enhanced net flow with stronger respiratory modulation. Flow directionality tests revealed no significant differences between forward and reverse flow under either cardiac- or respiratory-like conditions (Fig. S3). However, respiratory-driven conditions showed a slight tendency toward greater net flow in the forward direction, suggesting a mild directional bias when there is bulk transport. Together, these findings confirm that PC-MRI can reliably quantify respiration-related net flow, providing a robust basis for interpreting in vivo CSF measurements.

### CSF flow differences between trained and non-trained participants

We compared four CSF flow features (mean peak velocity, mean speed, displacement, and net flow) between T and NT participants at the FM and LV (Fig. 1). These features were chosen to capture differential cardiac- and respiratory-driven aspects of CSF movement, with FM representing the primary craniospinal exchange site and LV reflecting intraventricular movement in deeper brain regions. CSF mean peak velocity (cm/s) represents the average peak velocity of cardiac-driven pulsatile CSF waves, CSF mean speed (cm/s) reflects the average absolute velocity contribution of cardiac-driven pulsatile CSF waves, CSF displacement (ml) represents the total caudal–rostral movement of CSF during a respiratory cycle, and CSF net flow (µl) quantifies the respiratory-cycle-averaged directional bias in cranial versus caudal transport.

Group comparisons are summarized in Fig. 1. During RB at the FM, T showed higher CSF mean peak velocity (T 1.00 ± 0.24 cm/s; NT 0.78 ± 0.20 cm/s; $p = 0.0019$), higher CSF mean speed (T 0.83 ± 0.22 cm/s; NT 0.66 ± 0.18 cm/s; $p = 0.0059$) and higher CSF displacement (T 0.25 ± 0.20 ml; NT 0.09 ± 0.07 ml; $p = 0.0005$), with a trend toward higher CSF net flow (T 33.33 ± 42.41 µl; NT 15.46 ± 15.55 µl; $p = 0.0573$). During RB at the LV, T showed no difference in CSF mean peak velocity (T 0.21 ± 0.07 cm/s; NT 0.19 ± 0.05 cm/s; $p = 0.3238$) but higher values across the other three measures: CSF mean speed (T 0.19 ± 0.02 cm/s; NT 0.15 ± 0.04 cm/s; $p = 0.0411$), CSF displacement (T 0.049 ± 0.052 ml; NT 0.018 ± 0.014 ml; $p = 0.0093$), and CSF net flow (T 10.78 ± 12.01 µl; NT 4.25 ± 1.07 µl; $p = 0.0229$) (Fig. 1a). During DB at the FM, T showed higher values across all four measures: CSF mean peak velocity (T 1.02 ± 0.29 cm/s; NT 0.77 ± 0.23 cm/s; $p = 0.0022$), CSF mean speed (T 0.84 ± 0.27 cm/s; NT 0.61 ± 0.18 cm/s; $p = 0.0013$), CSF displacement (T 0.59 ± 0.32 ml; NT 0.22 ± 0.16 ml; $p < 0.0001$), and CSF net flow (T 70.28 ± 57.91 µl; NT 37.28 ± 29.59 µl; $p = 0.0172$). During DB at the LV, CSF mean peak velocity (T 0.21 ± 0.07 cm/s; NT 0.16 ± 0.05 cm/s; $p = 0.0135$) and CSF mean speed was significantly higher in T (T 0.19 ± 0.07 cm/s; NT 0.14 ± 0.04 cm/s; $p = 0.0176$), while CSF displacement and CSF net flow showed no group differences (Fig. 1b). Overall, CSF flow feature differences between T and NT were consistently observed at the FM across both breathing conditions, whereas in the LV they were more evident during RB.

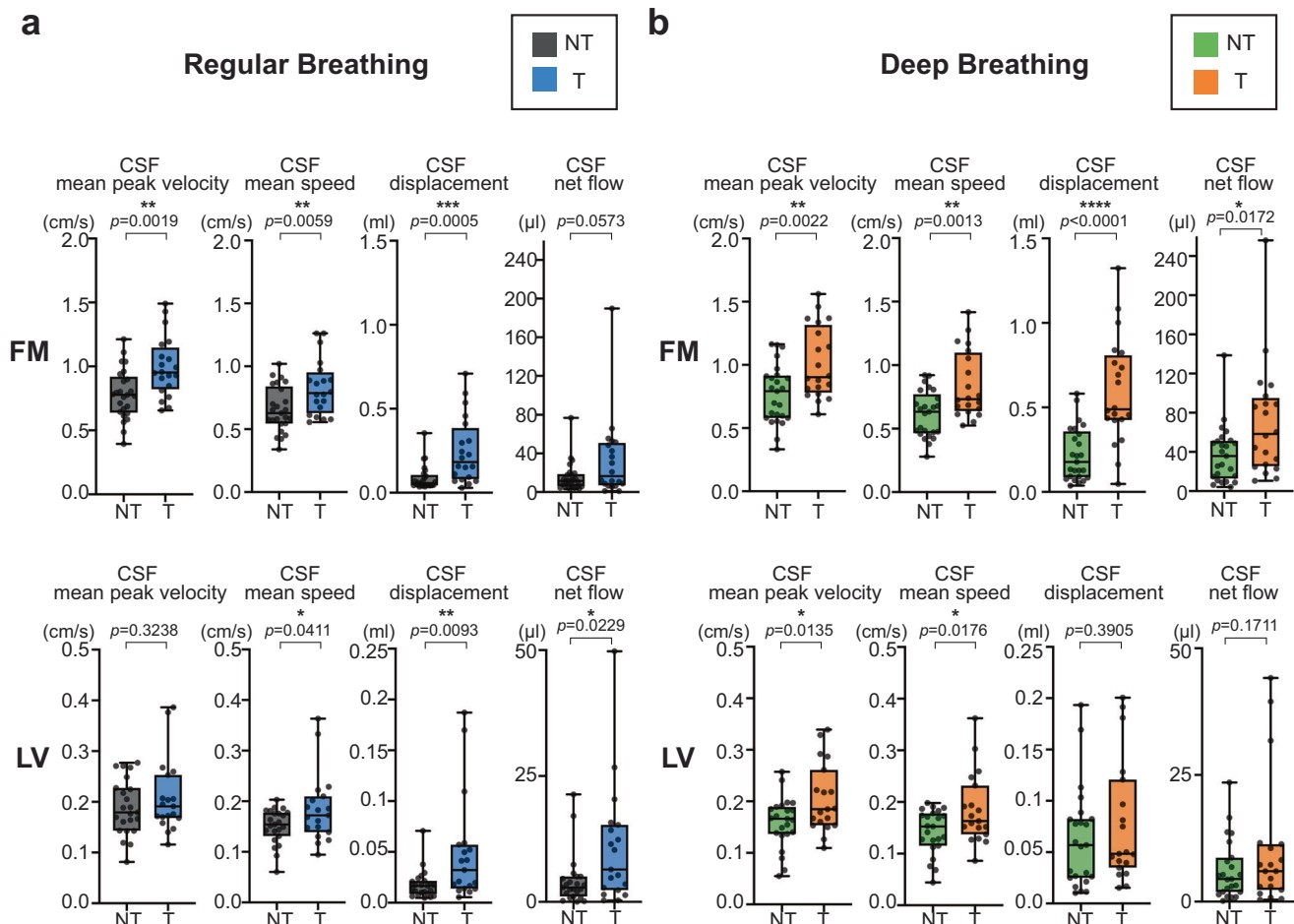

**Fig. 1 | CSF flow group differences between breathing practice–T and NT participants.** Groups were compared for CSF flow in the FM (T = 20, NT = 25) and LV (T = 19, NT = 23). Analyses included CSF mean peak velocity, CSF mean speed, CSF displacement, and CSF net flow were analyzed. Two-sided Student's t-tests were used to compare the NT and T groups. **a** During RB, group differences were observed in CSF mean peak velocity (*p* = 0.0019), CSF mean speed (*p* = 0.0059) and CSF displacement (*p* = 0.0005) in the FM. In the LV, group differences were observed in CSF mean speed (*p* = 0.0411), CSF displacement (*p* = 0.0093), and CSF net flow (*p* = 0.0229), while CSF mean peak velocity (*p* = 0.3238) did not differ between groups. **b** During DB, group differences were observed in all four measures in the FM: CSF mean peak velocity, *p* = 0.0022; CSF mean speed, *p* = 0.0013;

CSF displacement, *p* < 0.0001; CSF net flow, *p* = 0.0172. In the LV, group differences were observed only in CSF mean peak velocity (*p* = 0.0135) and CSF mean speed (*p* = 0.0176). In the whisker plots, NT during RB is shown in gray, T during RB in light blue, NT during DB in light green, and T during DB in orange for comparison. Box plots represent the 25th–75th percentile, whiskers indicate minimum and maximum values, and the central line shows the median. Statistical significance is denoted by asterisks as follows: *p* < 0.05 (*), *p* < 0.01 (**), *p* < 0.001 (***), and *p* < 0.0001 (****). No adjustments were applied for multiple comparisons. Source data are provided as a Source Data file. CSF cerebrospinal fluid, DB deep breathing, FM foramen magnum, LV lateral ventricle, NT non-trained, RB regular breathing, T trained.

---

Within-subject comparisons of RB and DB (Fig. S4) showed that CSF displacement increased with DB in both the FM and LV across T and NT participants (T-FM *p* = 0.0003; NT-FM *p* < 0.0001; T-LV *p* = 0.0470; NT-LV *p* < 0.0001), whereas CSF net flow increased with DB only in the FM (T-FM *p* = 0.0199; NT-FM *p* = 0.0024). In the LV NT, DB reduced the CSF mean peak velocity compared to RB (T-LV *p* = 0.6547; NT-LV *p* = 0.0005). To provide an overall comparison between groups and breathing conditions, we performed additional confidence interval–based analyses using NT-RB as the reference group (mean ± 95% CI) (Fig. S5). These analyses demonstrated that T consistently exhibited higher CSF flow features across both RB and DB compared with NT-RB. At the FM, CSF displacement was 0.087 ml (95% CI: 0.057–0.118) for NT-RB, 0.248 ml (0.155–0.340) for T-RB, and 0.594 ml (0.446–0.743) for T-DB, with both T-RB and T-DB significantly higher than NT-RB. At the FM, CSF net flow was 15.46 µl (95% CI: 9.05–21.88) for NT-RB, 33.33 µl (13.48–53.18) for T-RB, and 70.28 µl (43.18–97.38) for T-DB, with T-DB significantly higher than NT-RB. At

the LV, CSF displacement was 0.018 ml (95% CI: 0.012–0.024) for NT-RB, 0.049 ml (0.024–0.074) for T-RB, and 0.077 ml (0.049–0.106) for T-DB, with T-DB significantly elevated relative to NT-RB. At the LV, CSF net flow was 4.25 µl (95% CI: 2.03–6.46) for NT-RB, 10.78 µl (5.00–16.57) for T-RB, and 10.51 µl (4.20–16.82) for T-DB; although these differences were not statistically significant, values in T tended to be higher.

A sensitivity analysis was performed by regressing out RR (Fig. S6). CSF mean peak velocity and CSF mean speed were not affected by this adjustment. At the FM, the RB difference in the CSF displacement disappeared, while the DB difference between groups remained. At the LV, group differences in displacement and net flow were no longer significant, particularly during RB.

Among the representative peak-based velocity features (Fig. S7), during RB the FM showed group differences across all CSF velocity measures, including CSF mean peak velocity, CSF mean valley velocity, CSF mean peak-to-peak velocity, and CSF mean speed. In contrast, at the LV during RB, only CSF mean speed differed between NT and T.

During DB, all peak-based velocity features showed significant differences between NT and T at both the FM and LV. All peak-based velocity features showed very high correlations with one another ($p < 0.0001$) (Fig. S8).

Overall, when comparing T and NT, trained participants consistently exhibited higher CSF flow features across conditions. FM and LV showed different patterns, with T vs. NT differences evident at the FM in both RB and DB, but more apparent at the LV during RB. Among the CSF flow features, displacement was particularly sensitive to both training and DB. CSF flow features were not associated with years of training, suggesting that practice duration alone may not account for the observed group differences (Fig. S9). Even when LV displacement

did not differ between T and NT during DB, confidence interval analysis showed that LV displacement in both T and NT was higher than in NT-RB.

## Physiological features linking respiration to CSF Flow in FM and LV

To evaluate causal pathways and the unique contributions of respiratory and physiological variables to CSF dynamics, we applied structural equation modeling (SEM) supported by multivariate regression (Fig. 2a). We focused on two physiological features of interest: superior sagittal sinus (SSS) displacement, reflecting the mechanical component of the respiratory pump driving changes in venous return, and HR

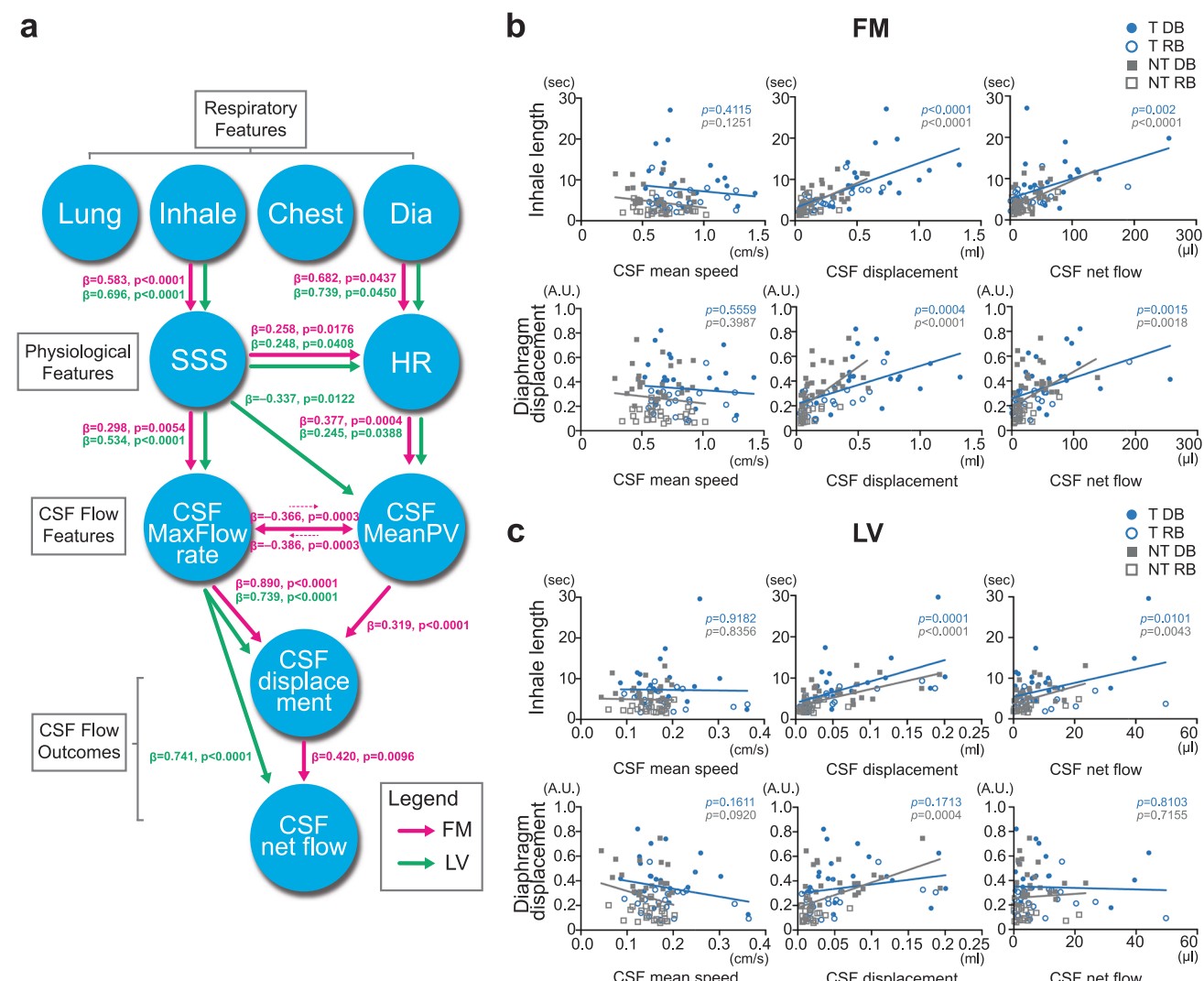

**Fig. 2 | Modeling the path from respiration to CSF net flow. a** To assess causality and evaluate unique contributions, we performed structural equation modeling and multivariate regression. Results indicated that in the FM (T = 20, NT = 25) CSF net flow is influenced by both mechanical (SSS) and autonomic (HR) pathways, whereas in the LV (T = 19, NT = 23) CSF dynamics were primarily driven by the mechanical SSS pathway. The lines represent β-values, which are standardized regression estimates from the multivariate analysis, with FM shown in pink and LV in light green. **b, c** Inhale length and diaphragm displacement, identified as key respiratory features from the SEM modeling, were examined for correlations with CSF mean speed, CSF displacement, and CSF net flow in the FM and LV. In the FM, both inhale length and diaphragm displacement strongly correlated with CSF displacement and CSF net flow. Specifically, inhale length correlated with CSF displacement (T: $p < 0.0001$; NT: $p < 0.0001$) and CSF net flow (T: $p = 0.002$; NT: $p < 0.0001$), as well as diaphragm displacement correlated with CSF displacement

(T: $p = 0.0004$; NT: $p < 0.0001$) and CSF net flow (T: $p = 0.0015$; NT: $p = 0.0018$). In the LV, inhale length correlated with CSF displacement (T: $p = 0.0001$; NT: $p < 0.0001$) and CSF net flow (T: $p = 0.0101$; NT: $p = 0.0043$), whereas diaphragm displacement correlated only with CSF displacement (T: $p = 0.1713$; NT: $p = 0.0004$). These correlations support the SEM modeling with FM and LV differential pathways linking respiratory features to CSF dynamics. Linear regression analyses were performed using Pearson's correlation with two-sided tests. Pearson's correlation analyses were conducted separately in T (light blue) and NT (gray). Source data are provided as a Source Data file. A.U. arbitrary unit, Chest chest displacement, CSF cerebrospinal fluid, DB deep breathing, Dia diaphragm displacement, Dis CSF displacement, FM foramen magnum, HR HR displacement, Lung lung area change during one respiratory cycle, LV lateral ventricle, MaxFlow rate CSF maximum flow rate, MeanPV mean difference between peak and valley velocity, NT non-trained, RB regular breathing, SSS superior sagittal sinus, T trained.

displacement, reflecting the autonomic component of respiratory modulation of cardiac inflow via respiratory sinus arrhythmia[27]. As a preliminary step, we confirmed that respiratory, physiological, and CSF flow features were highly correlated (Fig. S10), supporting the selection of these two variables as key mediators. Based on this rationale, we constructed a hierarchical SEM framework in which respiratory features were allowed to influence physiological variables, physiological variables to influence CSF flow variables, and CSF flow variables to influence one another.

At the FM, SEM indicated that respiratory influences acted through both a mechanical pathway, represented by SSS displacement, and an autonomic pathway, represented by HR displacement. Together, these pathways contributed to enhanced CSF displacement and net flow at the FM, demonstrating that both mechanical and autonomic mechanisms shape craniospinal fluid exchange. In contrast, in the LV, CSF dynamics were explained predominantly by the mechanical SSS pathway, with minimal contribution from autonomic effects (Fig. 2a).

Correlation analyses confirmed these relationships (Fig. 2b). A more detailed list of respiratory and CSF flow features is provided in Fig. S11. Respiratory cycle length (Fig. S11) had the strongest overall correlation with CSF flow measures (FM CSF mean speed: T $R^2 = 0.0168$, $p = 0.4259$; NT $R^2 = 0.025$, $p = 0.2766$; FM CSF displacement: T $R^2 = 0.4275$, $p < 0.0001$; NT $R^2 = 0.3722$, $p < 0.0001$; FM CSF net flow: T $R^2 = 0.2558$, $p = 0.0009$; NT $R^2 = 0.3005$, $p < 0.0001$; LV CSF mean speed: T $R^2 = 0.0005$, $p = 0.8961$; NT $R^2 = 0.0002$, $p = 0.7771$; LV CSF displacement: T $R^2 = 0.39$, $p < 0.0001$; NT $R^2 = 0.3851$, $p < 0.0001$; LV CSF net flow: T $R^2 = 0.1999$, $p = 0.0049$; NT $R^2 = 0.2027$, $p = 0.0017$). Within respiratory cycle length, inhale length and diaphragm displacement emerged as the primary drivers (Fig. 2a). At the FM, both inhale length and diaphragm displacement correlated strongly with CSF displacement and CSF net flow (Inhale length vs. FM CSF displacement: T $R^2 = 0.4066$, $p < 0.0001$; NT $R^2 = 0.4191$, $p < 0.0001$; Inhale length vs. FM CSF net flow: T $R^2 = 0.2256$, $p = 0.002$; NT $R^2 = 0.3705$, $p < 0.0001$; diaphragm displacement vs. FM CSF displacement: T $R^2 = 0.2821$, $p = 0.0004$; NT $R^2 = 0.3476$, $p < 0.0001$; diaphragm displacement vs. FM CSF net flow: T $R^2 = 0.2365$, $p = 0.0015$; NT $R^2 = 0.1846$, $p = 0.0018$) (Fig. 2b). At the LV, inhale length correlated with CSF displacement and CSF net flow, whereas diaphragm displacement correlated only with CSF displacement (Inhale length vs. LV CSF displacement: T $R^2 = 0.3368$, $p = 0.0001$; NT $R^2 = 0.3731$, $p < 0.0001$; Inhale length vs. LV CSF net flow: T $R^2 = 0.1699$, $p = 0.0101$; NT $R^2 = 0.1711$, $p = 0.0043$; diaphragm displacement vs. LV CSF displacement: T $R^2 = 0.0513$, $p = 0.1713$; NT $R^2 = 0.2499$, $p = 0.0004$) (Fig. 2c).

Direct correlations of physiological features with CSF flow measures showed a similar pattern as in the SEM results (Fig. S12). In the FM, SSS displacement correlated strongly with CSF displacement (T $R^2 = 0.2899$, $p = 0.0003$; NT $R^2 = 0.0837$, $p = 0.0416$), while HR displacement showed only a trend with CSF displacement only with T (T $R^2 = 0.0913$, $p = 0.0580$; NT $R^2 = 0.0458$, $p = 0.1356$) (Fig. S12a). In the LV, SSS displacement also correlated with CSF displacement (T $R^2 = 0.4181$, $p < 0.0001$; NT $R^2 = 0.0900$, $p = 0.0428$) and showed correlation with CSF net flow only with T (T $R^2 = 0.1823$, $p = 0.0075$; NT $R^2 = 0.0057$, $p = 0.6175$), whereas HR displacement did not correlate with any CSF flow features (Fig. S12b).

We also observed strong correlations at the FM between CSF net flow and CSF displacement (T $R^2 = 0.3041$, $p = 0.0002$; NT $R^2 = 0.2796$, $p < 0.0001$). At the LV, CSF net flow was instead correlated with CSF mean speed (T $R^2 = 0.1641$, $p = 0.0116$; NT $R^2 = 0.0898$, $p = 0.0431$) (Fig. S12). Together, this highlights regional specialization in the mechanisms driving CSF transport.

Exploratory joint SEMs were conducted for transparency. For the FM model, the joint SEM yielded $\chi^2(19) = 104.4$, CFI = 0.87, TLI = 0.69, RMSEA = 0.22, and SRMR = 0.16. For the LV model, the joint SEM yielded $\chi^2(19) = 75.7$, CFI = 0.90, TLI = 0.76, RMSEA = 0.19, and SRMR = 0.10. Although overall model fit was limited, the path estimates were consistent with those obtained from the hypothesis-driven hierarchical regression framework.

## Respiratory and Physiological Feature Differences Between Trained and Non-trained Groups

We examined T and NT group-level differences in respiratory and physiological features (Fig. 3). During RB, the T group showed significantly longer inhale length (T $5.046 \pm 2.264$ sec; NT $2.652 \pm 1.083$ sec; $p < 0.0001$), greater diaphragm displacement (T $0.2385 \pm 0.11$ A.U.; NT $0.1305 \pm 0.0537$ A.U.; $p < 0.0001$), and larger HR displacement (T $4.498 \pm 3.709$ ΔBPM; NT $2.806 \pm 1.521$ ΔBPM; $p = 0.0437$) and SSS displacement (T $0.0801 \pm 0.0969$ ml; NT $0.0339 \pm 0.0401$ ml; $p = 0.0359$) compared with the NT group. No difference is shown with chest displacement. During DB, these group differences diminished and were no longer statistically significant, except for inhale length (T $9.974 \pm 5.908$ sec; NT $6.673 \pm 2.499$ sec; $p = 0.0151$), which remained longer in the T group, and chest displacement (T $0.0266 \pm 0.0165$ A.U.; NT $0.0492 \pm 0.0243$ A.U.; $p = 0.0009$), for which the NT was larger. A more detailed list of respiratory and physiological features is provided in Fig. S13.

To illustrate these patterns at the individual level, representative cases from an NT and a T participant are shown (Fig. S14, 15). Each figure displays respiratory, cardiac, and FM CSF dynamics during RB and DB. Both participants belonged to the upper 50% group with high FM net flow. In both cases, spectral analyses highlight distinct respiratory and cardiac contributions. The NT participant (male, 62 yr) showed an increase in CSF mean speed ($0.59 \rightarrow 0.83$ cm/s), displacement ($0.10 \rightarrow 0.21$ ml), and net flow ($20.27 \rightarrow 64.72$ μl) during DB. The T participant (male, 59 yr) exhibited greater overall CSF dynamics, with mean speed ($0.89 \rightarrow 1.05$ cm/s) and displacement ($0.30 \rightarrow 1.32$ ml) increasing during DB, while net flow varied ($48.59 \rightarrow 24.64$ μl). These examples demonstrate the coupling between respiration and CSF flow in both T and NT individuals. Diaphragm movement videos from the same participant during RB and DB are also provided (Supplementary Videos 1–8). The playback speed is faster than real time for visualization purposes.

## Respiratory and Physiological Features Linked to Higher CSF Net Flow Across Participants

To identify respiratory and physiological features associated with stronger CSF movement, we ranked all participants by FM net flow and divided them at the median into a High group and a Low group (Fig. 4). This approach was applied without regard to breathing condition or training status, allowing us to capture generalizable features. At the FM, the High group included participants across both groups and breathing conditions (9 T-RB, 17 T-DB, 4 NT-RB, 15 NT-DB, out of 20 in T and 25 in NT for each breathing condition), and at the LV, the High group similarly included a mixture of participants (8 T-RB, 16 T-DB, 4 NT-RB, 13 NT-DB, out of 19 in T and 23 in NT for each breathing condition). Participants in the High group showed significantly longer inhale length (High $7.999 \pm 4.701$ sec; Low $3.857 \pm 2.161$ sec; $p < 0.0001$), greater chest displacement (High $0.0326 \pm 0.0238$ A.U.; Low $0.0242 \pm 0.0156$ A.U.; $p = 0.0497$), and greater diaphragm displacement (High $0.3935 \pm 0.1725$ A.U.; Low $0.2085 \pm 0.1330$ A.U.; $p < 0.0001$) compared with those in the Low. Among physiological measures, SSS displacement was higher in the High group (High $0.1908 \pm 0.1930$ ml; Low $0.0751 \pm 0.1785$ ml; $p = 0.004$), whereas HR displacement did not differ between groups. Higher FM net flow was also associated with greater FM CSF mean peak velocity (High $0.9347 \pm 0.2739$ cm/s; Low $0.8204 \pm 0.2354$ cm/s; $p = 0.0366$), FM CSF displacement (High $0.4088 \pm 0.2973$ ml; Low $0.1369 \pm 0.1386$ ml; $p < 0.0001$) and with increased LV CSF displacement (High $0.0721 \pm 0.057$ ml; Low $0.0304 \pm 0.0327$ ml; $p < 0.0001$) and net flow (High $9.809 \pm 10.02$ μl; Low $5.678 \pm 8.881$ μl; $p = 0.0486$). A more

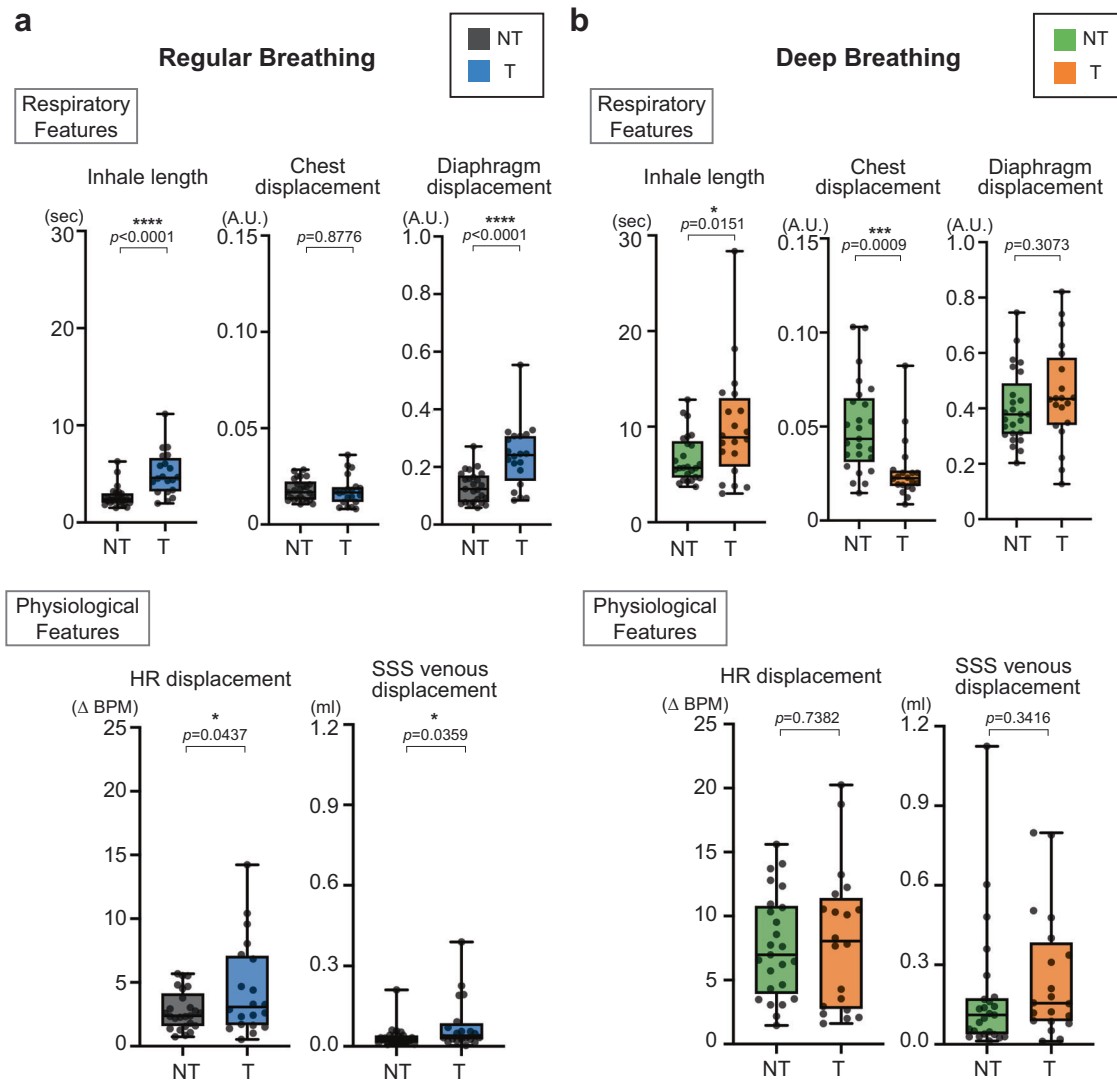

**Fig. 3 | Respiratory and physiological feature differences between breathing practice–T and NT groups.** Two-sided Student's t-tests were used to compare the NT and T groups. **a**, During RB, the T group showed higher inhale length ($p < 0.0001$) and diaphragm displacement ($p < 0.0001$) than the NT group, while chest displacement showed no difference between groups ($p = 0.8776$). HR displacement ($p = 0.0437$) and SSS venous displacement ($p = 0.0359$) were also greater in the T group compared with the NT group. **b** During DB, group differences diminished overall, with inhale length remaining longer in the T group ($p = 0.0151$). Chest displacement was larger in the NT group ($p = 0.0009$), whereas diaphragm displacement ($p = 0.3073$), HR displacement ($p = 0.7382$), and SSS venous displacement ($p = 0.3416$) showed no significant differences between

groups. Results are based on combined data from the FM (T = 20, NT = 25) and LV (T = 19, NT = 23). In the box plots, NT during RB is shown in gray, T during RB in light blue, NT during DB in light green, and T during DB in orange for comparison. Box plots represent the 25th–75th percentile, whiskers indicate minimum and maximum values, and the central line shows the median. Statistical significance is denoted by asterisks as follows: $p < 0.05$ (*), $p < 0.01$ (**), $p < 0.001$ (***), and $p < 0.0001$ (****). No adjustments were applied for multiple comparisons. Source data are provided as a Source Data file. A.U. arbitrary unit, BPM beats per minute, DB deep breathing, FM foramen magnum, HR heart rate, LV lateral ventricle, NT non-trained, RB regular breathing, SSS superior sagittal sinus, T trained.

detailed list of respiratory and physiological features is provided in Fig. S16. An extended set of respiratory features also differed between High and Low groups, including exhale length (High $4.147 \pm 1.712$ sec; Low $2.776 \pm 1.366$ sec; $p < 0.0001$), inhale-to-exhale ratio (High $2.002 \pm 0.9026$ A.U.; Low $1.438 \pm 0.4922$ A.U.; $p = 0.0004$), and lung area displacement (High $0.3744 \pm 0.1755$ A.U.; Low $0.1972 \pm 0.1229$ A.U.; $p < 0.0001$) (Fig. S16).

## Discussion

In this study, we found that trained participants consistently exhibited enhanced CSF dynamics compared with non-trained controls, though the patterns varied across brain regions and breathing conditions (Fig. 1). Causal inference modeling revealed that CSF flow at the FM was influenced by both mechanical (SSS) and autonomic (HR) pathways,

whereas LV flow was driven predominantly by mechanical effects (Fig. 2). These group-level differences were accompanied by longer inhale length and greater diaphragm displacement in trained participants, particularly during RB, reflecting temporal and special features of respiration that were closely coupled with CSF dynamics (Fig. 3). Extending beyond group comparisons, analysis across all participants showed that higher CSF net flow was linked to longer inhale length, greater diaphragm displacement, and higher SSS displacement, underscoring the importance of both temporal and mechanical aspects of respiration in driving CSF movement (Fig. 4).

We focused on the FM and LV because they represent two anatomically and functionally distinct compartments of CSF flow. The FM, located at the craniospinal interface, is a major conduit for cardiorespiratory pulsations driving CSF movement into the cranial cavity. In

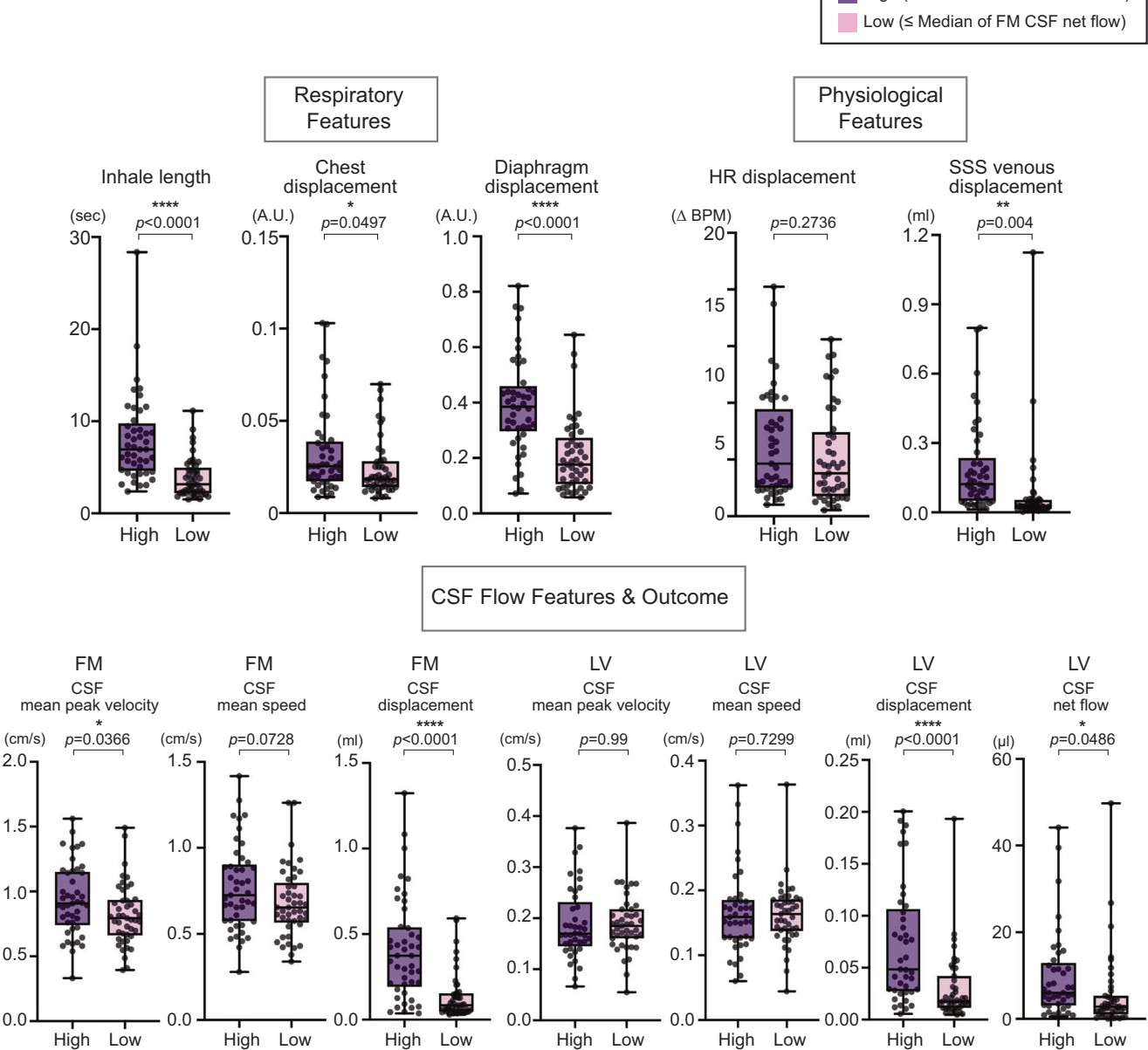

**Fig. 4 | Respiratory and physiological features related to increased CSF net flow.** To isolate features associated with CSF net flow, Participants were divided into High and Low groups based on the median of the FM CSF net flow. Group differences were assessed using two-sided Student's t-tests. The High group showed significantly greater inhale length ($p < 0.0001$), chest displacement ($p = 0.0497$), and diaphragm displacement ($p < 0.0001$). Among physiological features, SSS venous displacement was higher in the High group ($p = 0.0040$), whereas HR displacement showed no difference ($p = 0.2736$). Higher FM net flow was also associated with greater FM CSF displacement ($p < 0.0001$) and CSF mean peak velocity ($p = 0.0366$), while CSF mean speed did not differ significantly ($p = 0.0728$). Importantly, increased FM CSF net flow was linked to greater LV CSF displacement ($p < 0.0001$) and LV CSF net flow ($p = 0.0486$), whereas LV CSF mean peak velocity ($p = 0.99$) and mean speed ($p = 0.7299$) showed no difference between groups. Participant numbers (T = 19, NT = 23) reflect those who had both FM and LV data. In the box plots, the High group is shown in purple and the Low in pink for comparison. Box plots represent the 25th–75th percentile, whiskers indicate minimum and maximum values, and the central line shows the median. Statistical significance is denoted by asterisks as follows: $p < 0.05$ (*), $p < 0.01$ (**), $p < 0.001$ (***), and $p < 0.0001$ (****). No adjustments were applied for multiple comparisons. Source data are provided as a Source Data file. BPM beats per minute, CSF cerebrospinal fluid, FM foramen magnum, HR heart rate, LV lateral ventricle, NT non-trained, SSS superior sagittal sinus, T trained.

contrast, the LV is a deeper brain region that has received less attention in CSF research, despite being the primary site of CSF production via the choroid plexus and its structural connections to memory-related regions such as the hippocampus and entorhinal cortex[28,29]. The LV has been reported to have less cardiac and respiratory pulsatility[30,31], and our results showing a respiratory effect provide important evidence that these influences extend beyond the craniospinal interface into deeper brain structures.

These findings align with the model results, indicating that LV dynamics are primarily driven by respiration, whereas FM reflects both respiratory and additional autonomic influences. The RR regression analysis highlights the importance of respiratory rate but also shows that FM differences during deep breathing persist, suggesting involvement of mechanisms beyond simple respiratory modulation. Additionally, the T group exhibited a slower respiratory rate even during regular breathing, supporting a training-related adaptation rather than

an intentional effort to breathe slowly. Participants were instructed to maintain a natural and sustainable breathing rhythm without consciously controlling their rate. This distinction is important, as untrained attempts to reduce respiratory rate can elevate $CO_2$ levels and induce irregular or unstable respiration[32]. The observed effects, therefore, reflect a trained, physiologically integrated breathing pattern, rather than simply slower breathing.

A difference in CSF mean speed was observed between the T and NT groups, whereas no difference was found between breathing conditions within each group. This likely reflects that CSF mean speed primarily represents cardiac-driven CSF movements. Liu et al. reported that during deep breathing compared with free breathing, cardiac-driven CSF flow did not differ significantly, while respiration-driven CSF flow increased markedly (by up to 326%), indicating an enhanced respiratory contribution to overall CSF dynamics[12]. Similarly, Yildiz et al. showed that deep abdominal breathing increased the respiratory CSF component but not cardiac components[11], and Dreha-Kulaczewski et al. observed a strong inspiration effect when cardiac signals were removed[33]. Together, these findings support that CSF mean speed largely reflects the cardiac component, while displacement and net flow capture respiratory modulation. The higher baseline CSF mean speed in the trained group may relate to long-term breathing training effects on autonomic or vascular tone[34], as supported by our finding of greater HR displacement (RSA) during RB. Future studies including HRV and cerebrovascular reactivity are needed to clarify this mechanism[35].

In our analysis, HR displacement represents respiratory sinus arrhythmia (RSA), a respiration-driven modulation of HR that reflects sympatho–vagal balance[36,37]. While respiratory-induced venous return has occasionally been studied in CSF dynamics[14,31,38–40], the role of RSA in relation to CSF flow has been less explored. At the FM, the presence of a respiration-driven cardiac component in CSF dynamics supports a cardiorespiratory interaction (Fig. 2a). In our data, HR displacement correlated with all breathing features, highlighting a close connection between breathing mechanics and RSA (Fig. S10). This association was already evident in the T group during RB (Fig. 3), suggesting that long-term practice may help maintain cardiorespiratory coupling reflected in RSA.

Previous studies suggest that autonomic activity may influence glymphatic function, with sympathetic tone suppressing and parasympathetic tone enhancing glymphatic flow[41,42]. While we did not directly measure glymphatic clearance, our results demonstrate that respiration modulates CSF dynamics during wakefulness, which may provide a physiological context for future studies of glymphatic function. Consistent with this, a recent study demonstrated that enhancing RSA through paced breathing reduced plasma amyloid-β and tau levels in both younger and older adults, suggesting that autonomic regulation of respiration may contribute to lowering neurodegenerative risk[43].

Our results show that CSF displacement, rather than net flow, better reflects physiological influences on CSF dynamics. Displacement at the FM also emerged as the key intermediate feature leading to net flow. This suggests that when CSF movement occurs slowly but with a large magnitude, the balance between inflow and outflow is more likely to be uneven, resulting in a temporary net shift. Although the Monro–Kellie doctrine is typically applied to describe intracranial volume compensation under normal and chronic conditions[44], our findings indicate that its principles may also apply on smaller and more transient timescales. The doctrine describes the dynamic volume balance among brain tissue, blood, and CSF within the fixed cranial cavity[44–46]. In our dataset, raw net flow in either the positive or negative direction did not correlate with other variables (data not shown), so we focused on absolute values in relation to CSF mean speed and CSF displacement. One possible explanation is that the direction of net flow varies across individuals or over time. Taken together, these

findings imply that large respiration-induced changes in cardiac input and venous return can displace CSF and temporarily alter this intracranial balance.

Accurately measuring net CSF flow remains challenging. Most prior studies have emphasized oscillatory peak velocities rather than net displacement, and although net flow has been reported, few investigations provide comprehensive measurements over extended durations[9,47]. Importantly, net flow derived from 2D PC-MRI may not fully represent the true bulk circulation, as the technique measures velocity across a single imaging plane rather than whole-brain transport. To address potential technical offsets inherent to MRI acquisition, including low-frequency phase drift and gradient calibration shifts, we applied detrending to the CSF data prior to secondary analysis. Our phantom experiments further demonstrated good reproducibility under controlled conditions. To minimize variability in ROI placement, we implemented a semi-automatic procedure to make region identification more objective. In our analysis, detrending and segmentation by respiratory cycles further constrain the metric to reflect only the cardiorespiratory-driven component of CSF movement, rather than long-term circulation. Prior human studies have described a cranial CSF displacement of $1.0 \pm 0.9$ ml after 20 seconds of forced breathing[9], as well as slight cranial shifts in the aqueduct, L3, and especially C3 during forced breathing[33]. Complementary evidence from glymphatic research shows CSF mean flow speeds of ~30 µm/s in the perivascular space using tracer techniques[7], and modeling in mice has estimated rates of ~16.3 µm/s[48].

Directionality is an important aspect of CSF dynamics. Previous studies have reported a general pattern, with CSF moving rostrally during inhalation and caudally during exhalation[12,22], and have shown that this pattern can vary across anatomical regions of the spine and cranial base[18]. In the present study, we focused on the magnitude of CSF motion rather than its directionality. Amplitude-based measures were used to characterize overall CSF motion and its cardiorespiratory influences using 2D real-time PC-MRI. Future studies using 4D flow MRI could provide a more comprehensive assessment of respiratory effects on CSF directionality[31].

Our findings suggest that breathing training helps preserve or enhance respiratory influence on CSF dynamics, a potential physiological pathway that may offset declines related to aging or disease. Prior studies have shown that CSF velocities in the FM and ventricular system decline with age, likely due to anatomical changes and arterial stiffening, which reduce the transmission of intravascular forces to the CSF[49–51]. Conditions such as hypertension and reduced venous compliance can further impair CSF dynamics by stiffening vessels and weakening pressure-driven drainage[52,53]. Both hypertension and heart failure have also been linked to reduced glymphatic function[54–56]. Our result also suggests that the respiratory effect could influence CSF movement even in deeper brain regions like LV. As current clinical strategies to modulate CSF flow are primarily invasive, and there is emerging interest in applying noninvasive approaches[57], these results position respiratory training as a feasible and promising intervention to support macroscopic CSF dynamics, especially in individuals with compromised cardiovascular function. Future studies will be needed to clarify how respiration-driven CSF movements relate to glymphatic function, and dementia-focused investigations are planned to determine whether these effects translate into clinical benefit.

During RB, T participants showed greater diaphragm displacement compared with NT participants, while chest displacement did not differ between groups. Lung volume displacement was also higher in the T group. During DB, trained participants exhibited minimal chest wall movement with greater diaphragm excursion, whereas NT participants showed greater chest motion and less diaphragm contribution. The diaphragm displacement during DB did not reach statistical significance between groups ($p = 0.3073$), and no difference was observed in lung volume displacement. These results indicate that T

participants primarily relied on diaphragm motion, whereas NT participants exhibited a more thoracic breathing pattern (Fig. S13).

While most individuals do not engage in breathing practice during daily life, involuntary or semi-voluntary actions such as sighing, hiccupping, coughing, yawning, sneezing, and laughing involve forceful diaphragmatic movements and prolonged breathing patterns that generate substantial swings in intrathoracic pressure. These brief but intense events have been shown to produce large pressure fluctuations capable of driving CSF movement, reinforcing the mechanical coupling between diaphragmatic motion and CSF flow[16,33,39]. Although the cumulative effect of these episodes on CSF circulation, or waste clearance remains uncertain, their magnitude suggests that diaphragm-driven forces have long been contributing to brain fluid homeostasis as part of everyday physiology, even without us being consciously aware.

This study has several limitations. First, non-trained participants were defined broadly, without considering shorter-term structured breathing programs, which have been shown to modulate physiological outcomes[58]. Second, the modeling showed a suboptimal global fit in the exploratory joint SEM (CFI < 0.90; RMSEA > 0.10), limiting the strength of causal interpretation even though the directional trends were consistent. Third, we did not include direct autonomic or cardiovascular measurements such as real-time BP, HR variability (HRV), or biochemical markers (catecholamines, cortisol), limiting interpretation of autonomic contributions. Future studies incorporating multimodal monitoring across autonomic conditions (e.g., exercise, sleep, pharmacologic interventions) will be needed to validate and extend these mechanisms, building on prior work showing coordinated neural, hemodynamic, and CSF oscillations during sleep and wakefulness in the human brain[59,60]. Another possible limitation of this study is that trained participants, who have long-term experience with controlled breathing, may have intentionally or unintentionally modulated their respiration during scanning, which could have influenced group differences. Although all participants received identical instructions and followed the same standardized protocol, minor differences in breathing behavior cannot be entirely ruled out. In addition, breathing patterns during MRI may not fully reflect the practitioners' natural breathing patterns due to the confined scanning environment and physical constraints. Respiratory signals were recorded using a pressure-balloon belt with a fixation strap positioned on the upper abdomen just below the sternum, which may have restricted full chest and abdominal expansion and slightly altered breathing depth or rhythm. These factors should be considered when interpreting the findings, and future studies using blinded designs and less restrictive setups may help better isolate true training-related effects.

In conclusion, our findings highlight the active role of respiration in modulating CSF dynamics in the awake state. Seokmun Hoheup practitioners exhibited greater CSF modulation even during RB. Breathing influences CSF net flow through autonomic effects on cardiac activity (input) and mechanical effects on venous return (output), consistent with the Monro-Kellie principle of intracranial balance. This simple yet delicately coordinated mechanism, in which respiratory rhythm influences fluid dynamics in the brain, may be viewed as a form of 'respiration technology'. Respiration in the awake state emerges as a modifiable, noninvasive mechanism that supports involuntary functions such as CSF dynamics and may serve as a lifestyle intervention to promote brain health, with potential future applications in dementia prevention and care.

## Methods
### Flow phantom validation
A precision flow phantom test[61] was conducted to validate the accuracy of 2D PC-MRI performed in this study. To mimic the biomechanical environment of CSF, a custom upper torso phantom was used, filled with polyacrylic hydrogel matrix[62]. Specifically, 52.8 g of poly

(acrylic acid) partial sodium salt (Sigma-Aldrich, St. Louis, MO) and 26.4 g of sodium chloride (Sigma-Aldrich, St. Louis, MO) were dissolved in 30 L of deionized water during polymerization. A cylindrical tube (0.5-inch inner diameter) was embedded longitudinally within the gel (Fig. S17a), enabling controlled fluid transport through the tube. Flow velocity was regulated by a computer-controlled motor controller (Arcus Technology, The Colony, TX) driving a stepper motor (Vexta; Oriental Motor, Chicago, IL). The shielded motor and gear pump (Liquiflo, Garwood, NJ) were located in the MRI scan room (Fig. S17b). Flow rates were empirically calibrated against actual volumetric flow measurements and were reproducible to within ±0.1 cm/s. This design allowed the imposed pulsatile flow to closely approximate CSF dynamics in the brain.

Two tests were performed to validate the PC-MRI measurements. The first test mimicked cardiac-driven and respiration-driven pulsation of CSF flow by imposing unidirectional flow through the embedded tube to evaluate the accuracy of the PC-MRI flow analysis (Fig. S17a). The second test was conducted to assess potential baseline phase offset or drift in the MRI scanner system that would be employed for the in vivo study. This test examined the influence of flow directionality on the measured PC-MRI signal by alternating the flow direction while maintaining identical velocity and frequency parameters (Fig. S17a).

In the first test, six sessions were conducted by driving the motor system at sinusoidal velocities of 1, 2, and 3 cm/s with a cycle length of 1 s (cardiac pulsation), and 1.5, 2, and 2.5 cm/s with a cycle length of 8 s (respiratory pulsation). As shown in Fig. S17d, each of the six sessions was repeated twice with a 2 min 30 s interval, followed by a 2 min 30 s no-flow period. This entire sequence was repeated once more. It should be noted that in the cardiac pulsation conditions at 2 cm/s and 3 cm/s, the signals in the second and third cycles showed distortion artifacts and were excluded from subsequent analysis.

In the second test, four sessions were conducted at a fixed velocity of 2 cm/s, alternating between forward and reverse flow directions. Two conditions simulated cardiac pulsation (cycle length = 1 s) and two simulated respiratory pulsation (cycle length = 8 s). Each session was repeated three times (Fig. S3a,b).

Signal processing and quantitative analysis followed procedures identical to those used for the in vivo analysis, as described in the Methods, Data Processing and Analysis part.

### Participants
We recruited healthy participants, including 20 breathing practice-T individuals and 25 NT individuals (See Fig. S1, and Table S1). All participants provided written informed consent, and the study was approved by the Mayo Clinic Institutional Review Board. Inclusion criteria required participants to be healthy adults without medical conditions affecting brain function, concentration, memory, balance, or coordination. Exclusion criteria included MRI-incompatible implants, need for sedation, and pregnancy. Age and sex distributions were recorded and summarized in Table S1.

All were screened for prior breathing-related training history. The T group included individuals with sustained formal training in the breathing practice of Seokmun Hoheup[26], a lower belly-centered breathing method, each with more than one year of continuous practice. Participants with prior experience in other breathing practices were eligible for the T group, provided that Seokmun Hoheup had been their primary and dominant practice for at least the past year. For recruitment of the T group, we collaborated with Seokmun Domun, a non-profit organization, to identify and recommend practitioners with more than one year of consistent training (at least one session per week, approximately 90 minutes per session) who were interested in participating in the study. The NT group consisted of individuals with no formal sustained breathing practice longer than six months, confirmed by screening to exclude any prior structured

training such as diaphragmatic breathing, yoga, or mindfulness. The participant sample included four authors of the study, but all data were anonymized and analyzed by an independent researcher to ensure objectivity and minimize bias.

## Breathing Practice Background (T group only)

Participants in the T group engaged in Seokmun Hoheup practice, a structured breathing method designed to promote natural, lower belly-centered respiration[26]. The program is developed to support self-development and self-completion through a progressive, whole-body approach that cultivates lasting changes in breathing patterns. It combines coordinated physical movements with focused attention on a specific point in the lower abdomen to encourage natural and sustained breathing habits.

Each training session lasts approximately 90 minutes and consists of four components: a warm-up exercise (Chejo, 10 min), Haenggong (24 min), breathing practice (48 min), and recovery stretching (Hoegeonsul, 12 min). The warm-up exercise involves gentle body movements to stabilize physiological functions, relax muscles, and prepare the body and mind for the practice. Haenggong comprises 12 whole-body postures, each maintained for two minutes, to facilitate transition into the core breathing practice. The breathing practice is performed in different positions (lying down, sitting, or standing) depending on the training stage. Sessions conclude with recovery stretching, which consists of gentle movements synchronized with breathing to promote relaxation and improve circulation.

Through repeated experiential training, Seokmun Hoheup practice emphasizes body awareness, postural stability, and activation of the diaphragm, fostering fine, long, and deep breathing with an inhale-to-exhale ratio typically around 6:4[26]. In this study, participants in the T group performed the breathing practice component during the DB task of the MRI scan, consisting of natural, lower belly-centered slow and deep breathing. The practice does not involve any pacing or counting but emphasizes relaxed, natural breathing with focused attention on a specific point in the lower abdomen.

## Breathing task during MRI (Both NT and T groups)

All participants underwent PC-MRI and diaphragm scans under two breathing conditions: RB and DB. During the RB, participants followed their natural respiratory pattern. During the DB, participants were asked to breathe slowly and deeply while avoiding hyperventilation or breath-holding. The task conditions were alternated across participants, in order to reduce potential order effects. Each breathing state was performed in a free-breathing manner without any externally imposed pacing. The main instruction for both conditions was to engage in a breathing pattern that felt natural and sustainable. Study staff checked in between scans (every ~two minutes) to ensure compliance, keep participants alert, and monitor comfort and signs of drowsiness.

## Physiological measurement

During MRI scanning, respiratory signals were recorded using a respiratory measurement system with a pressure-balloon belt and a fixated strap, positioned on the upper abdomen just below the sternum. Peripheral pulse waveforms were simultaneously acquired using photoplethysmography (PPG). BP values were obtained separately from participants' medical records within an 8-month timeframe of the imaging session.

## MRI data acquisition

**Anatomical imaging.** MRI was performed on a 3 T Ingenia Elition X scanner (Philips Healthcare, Gainesville, Florida, USA) using a previously established protocol[17]. Each MR exam consists of three different imaging sessions: anatomical imaging, CSF flow acquisition in brain regions during the breathing task, and diaphragm imaging. A 20-channel head and neck coil was used for brain imaging and a 32-channel body coil for diaphragm imaging. Baseline anatomical imaging for the brain was performed with magnetization-prepared rapid gradient-echo (MPRAGE) and T2-weighted sequences. T1-weighted 3D sagittal MPRAGE images were acquired with the following parameters: long repetition time (TR) / inversion time (TI) / short TR / echo time (TE) = 1740 / 865 / 6.6 / 3.0 ms, flip angle (FA) = 9°, field of view (FOV) = $254 \times 254 \times 211$ mm³, voxel size = $1.0 \times 1.0 \times 1.0$ mm³, sensitivity encoding (SENSE) factors = $1 \times 2$, and total scan time = 6 m 11 s. T2-weighted 3D sagittal turbo-spin-echo (TSE) images were acquired with the following parameters: TR / TE = 1500/200 ms, flip angle (FA) = 9°, field of view = $228 \times 228 \times 36$ mm³, voxel size = $0.8 \times 0.8 \times 0.8$ mm³, TSE factor = 56, no acceleration factor, and total scan time = 6 m 6 s.

**CSF and Venous Flow measurements.** Flow measurements were performed using real-time PC-MRI with the following parameters: TR/TE = 102/46 ms, FA = 30°, field of view = $217 \times 217$ mm², matrix size = $172 \times 169$, in-plane resolution = $1.26 \times 1.28$ mm², slice thickness = 3 mm, SENSE factor = 3, temporal resolution = 0.19 s, and total scan time = 140 s. At each site, acquisition began at the lower velocity encoding (VENC) and was repeated at higher values if aliasing was observed. Brain regions for CSF flow measurements include the FM (VENC 5–10 cm/s; T $n = 20$, NT $n = 25$) and LV (VENC 2–5 cm/s; T $n = 19$, NT $n = 23$). Venous flow was also measured at the SSS (VENC 35–45 cm/s; T $n = 20$, NT $n = 25$). The slice plane was positioned perpendicular to the local flow direction. LV data were missing for three participants: one T (LV protocol not acquired early in the study), and two NT (incorrect slice location and excessive distortion). Additional exploratory acquisitions (e.g., 4th ventricle, aqueduct, C2/C3) were performed early in the study but were not included in the analyses reported here.

**Diaphragm motion imaging.** T2-weighted single-shot spin echo images were acquired in a sagittal orientation focused on the apex of the right lung. Slice position was guided by localizer imaging and positioned laterally to major vessels to minimize artifacts. Imaging parameters included: TR/TE = 857/120 ms, FA = 90°, in-plane resolution = $0.94 \times 0.94$ mm², and temporal resolution = 0.857 s.

## Data processing and analysis

**Feature extraction and quantification.** For the PC-MRI flow analysis, we followed the semi-automated approach described in a previous study[12] (Fig. S18 and S19). The analysis includes region of interest (ROI) selection, data extraction and preprocessing, and feature calculation. In this manuscript, we use the term CSF movement to refer to the velocity-derived local measures (mean speed, displacement, and net flow), and we use CSF dynamics to encompass the broader concept of cerebrospinal fluid behavior.

ROI selection: Each participant underwent six PC-MRI scans, FM, LV, and SSS regions, during both RB and DB. For each brain region, a bounding box was redefined to center the target structure and reduced the image size: $64 \times 64$ for FM and $32 \times 32$ for LV and SSS. Within this redefined bounding box, voxel-wise fast Fourier transform (FFT) analysis was applied to identify voxels showing cardiac-driven CSF oscillations (0.8 − 2.0 Hz). The relative spectral intensity ($I_{pi}$) was calculated as[63]:

$$I_{pi} = \frac{\int_{f_{\max\_i} - \Delta f}^{f_{\max\_i} + \Delta f} Pi(F)}{\int_{0.8}^{2} Pi(F)} \tag{1}$$

$\Delta f$ represents the frequency resolution (1/total acquisition time). Voxels showing strong cardiac-driven CSF signals were retained, and the resulting $I_{pi}$ maps were denoised and thresholded at the 95th percentile, followed by visual quality control (Fig. S18). To reduce noise, only voxels within ±0.2 Hz of the subject-specific cardiac

frequency peak were retained; others were set to zero[12]. For SSS, a similar approach was applied, however, because the cardiac-driven signal was weaker and respiration had a stronger influence[9,64,65], voxel selection was centered on the respiratory band (0.01–0.5 Hz) with a narrower tolerance of ±0.025 Hz. This final voxel mask derived from this process was used as the ROI.

Data extraction and preprocessing: The ROI mask was applied back to the raw PC-MRI image to extract the velocity time course. No frequency filtering was applied; the velocity signal was analyzed in its original form. As preprocessing, the velocity time course was detrended using a first-order (linear) polynomial to remove DC offsets and linear drift (MATLAB).

Feature calculation: The flow analysis progressed from velocity (cm/s) to flow rate (ml/s) and finally to volume (ml), with specific flow features extracted at each stage. The calculated CSF flow features included CSF mean peak velocity, CSF mean valley velocity, CSF mean PV (mean of peak and valley velocity), CSF mean speed (mean of the absolute CSF velocity), CSF maximum flow rate, CSF minimum flow rate, CSF displacement, and CSF net flow, along with the physiological SSS venous features. A summary of these flow features is provided in Table S2.

Peak detection was applied to the ROI derived velocity time course to identify peaks and valleys corresponding to the CSF systolic and diastolic peaks (MATLAB). A visual inspection was performed to verify the accuracy of peak detection against the simultaneously recorded PPG signal. The mean of all detected peaks and valleys within a scan yielded the CSF mean peak velocity (cm/s) and CSF mean valley velocity (cm/s), respectively. The average of the CSF mean peak velocity (cm/s) and CSF mean valley velocity (cm/s) provides CSF mean PV. The difference between the mean peak and mean valley velocities was calculated to obtain the CSF mean peak-to-peak velocity (cm/s), representing the amplitude of cyclic CSF velocity oscillations within each scan. To quantify overall CSF movement independent of direction, the absolute value of the velocity time course was averaged to obtain CSF mean speed[17].

To normalize for ROI size differences across scans, velocity (cm/s) was converted to volumetric flow rate (ml/s) by multiplying the mean velocity by voxel area, slice thickness, and voxel count, generating a CSF flow rate time course (Fig. S14). To further estimate the actual flow volume movement, the CSF flow rate time course was integrated using Newton–Cotes 5-point weights (Boole's rule) after tenfold spline upsampling, yielding the CSF volume displacement time course (ml)[12]. Representative examples are shown in Figs. S14 and S15.

A simultaneously recorded respiratory belt signal was used to segment the data into respiratory cycles by detecting valleys (local minima) (MATLAB). Each cycle was temporally normalized to 0–100% and interpolated to 200 points. The CSF flow rate time course and CSF volume displacement time course were resampled to this grid for breath-by-breath analysis. The maximum and minimum flow rates within each respiratory cycle were defined as CSF maximum flow rate (ml/s) and CSF minimum flow rate (ml/s), respectively. From the CSF volume displacement time course, CSF displacement (ml) was defined as the difference between maximum and minimum displacement within a respiratory cycle, and CSF net flow (μl) as the absolute sum of this maximum and minimum displacement. Displacement represents total CSF movement per respiratory cycle, while net flow quantifies the directional transport of CSF within each cycle (Fig. S14).

For the SSS venous flow analysis, the same processing steps were applied. The velocity (cm/s) time course was first extracted from the ROI, then converted to flow rate (ml/s), and subsequently integrated to obtain the volume displacement (ml) time course. Using the flow rate time course segmented by respiratory cycles, the SSS venous max flow rate was determined. When a bandpass filter was applied within the respiratory frequency range (0.01–0.5 Hz), the SSS venous max flow rate (Resp) was obtained. From the volume displacement time course

(ml) aligned with the respiratory cycle, the SSS venous displacement was calculated as the total oscillatory volume change from maximum to minimum within a respiratory cycle. Applying a bandpass filter in either the respiratory or cardiac frequency ranges yielded the SSS venous displacement (Resp) and the SSS venous displacement (Cardio), respectively. The bandpass filter parameters were determined based on subject-specific respiratory and cardiac frequencies derived from the respiratory belt and PPG signals (MATLAB).

To calculate HR displacement, the PPG signal was processed by detecting peaks, calculating inter-beat intervals, and converting them to beats per minute (BPM) to obtain HR. Within each respiratory cycle, HR change (ΔBPM) was defined as the difference between the maximum and minimum HR, yielding HR displacement. These metrics were then used to examine relationships between autonomic modulation and CSF dynamics.

From the respiratory belt data, respiratory features including inhale length, exhale length, and RR were calculated. Additional features such as lung area displacement, chest displacement, and diaphragm displacement were obtained through the following Diaphragm Motion Analysis.

**Diaphragm motion analysis.** Diaphragmatic motion during respiration was quantified using DeepLabCut (version 3.0.0rc6), a deep learning framework[66] for markerless pose estimation. Eight anatomical landmarks were manually defined in the DeepLabCut graphical user interface (GUI), as shown in Fig. S19d and S20. For model initialization, training frames were extracted from the diaphragm cross-sectional MP4 videos using the built-in Extract Frames function with default settings (automatic extraction, k-means clustering, no cropping).

Model training and inference were performed using PyTorch on a workstation equipped with an NVIDIA GeForce RTX 4070 SUPER GPU. All training parameters were maintained at DeepLabCut's default settings, with the network architecture based on ResNet-50, initialized through transfer learning using ImageNet weights. Data augmentation was executed using the Albumentations library. Training was conducted over 1000 iterations, with a maximum of 200 epochs, and model checkpoints were saved every 50 epochs. The model was trained independently on 90 video recordings from 45 participants under both RB and DB conditions ($n = 90$), achieving an averaged test root-mean-square error (RMSE) of approximately 1.5 pixels (see Fig. S20). The trained model was then used to extract landmark coordinates from all videos, which were exported as CSV files for further analysis.

Based on the tracked landmarks, six motion metrics were defined (Fig. S20):
- Lung area represents the 2D cross-sectional area of the lung region enclosed by respiratory motion in each frame.
- Upper chest and lower chest motion are defined as the Euclidean distance between landmark pairs on the upper and lower thoracic boundary, respectively.
- Diaphragm 1, 2, and 3 are also defined as Euclidean distances between specific diaphragm-related landmarks.

These metrics allowed for a detailed characterization of diaphragmatic movement patterns across different respiratory phases. To account for inter-individual variability, all motion metrics were normalized to each subject's exhale baseline during RB. The baseline was defined as the square root of the mean cross-sectional lung area at local minima (valley points). The same normalization factor was applied to DB metrics.

Three primary summary parameters were derived from the motion analysis: lung area displacement, chest displacement, and diaphragm displacement. Because respiratory motion, including diaphragm activity, typically exhibits oscillatory sinusoidal dynamics, each parameter was calculated from the normalized time-series signal.

Lung area displacement represented the total change in the 2D cross-sectional lung area across the respiratory cycle. Chest displacement was quantified by calculating the upper envelope of the normalized chest motion signal and taking the mean difference, representing the total oscillation amplitude. Diaphragm displacement was computed using the same approach. Among the three diaphragm landmarks, diaphragm 3, located in the posterior region, was used, as this region has been reported to exhibit the greatest excursion during respiration[67]. Because our imaging plane was positioned slightly off-center near the right lung apex, the diaphragmatic muscle anchored around the L1–L3 vertebral levels exhibited prominent downward motion during inspiration, reflecting posterior diaphragmatic movement. To accurately quantify this effect, we marked the posterior diaphragmatic contour along the dome and the diaphragm sulci connecting to the chest and back walls.

**Feature selection and causality modeling.** We used a two-step modeling approach[68,69] combining multivariate regression and SEM to examine directional relationships among respiratory, physiological, and CSF flow features. As for feature selection, we confirmed that most respiratory, physiological, and CSF variables were highly correlated with each other (Fig. S10). A multicollinearity check was performed using the variance inflation factor (VIF). Within each feature group, when variables showed high VIF values, the one with the strongest overall correlation with other features within the hierarchical structure of the model was retained. Inhale and exhale length showed extremely high VIF values (VIF = infinite), so inhale length was retained. Among CSF mean peak velocity, mean valley velocity, and mean PV, CSF mean PV was retained due to very high VIF values (VIF > $10^8$) among the three variables. Between CSF maximum and minimum flow, CSF maximum flow rate was retained (VIF > 7). The final model included inhale length, lung area displacement, chest displacement, and diaphragm displacement as respiratory features; SSS venous displacement and HR displacement as physiological features; and CSF mean PV, CSF maximum flow rate, CSF displacement, and CSF net flow as CSF flow features. CSF mean speed was excluded from the model as it showed no statistically significant correlation with any of the physiological features.

To ensure interpretability, we imposed a hierarchical assumption: respiratory features could influence only physiological variables, physiological features could influence only CSF flow variables, and CSF flow variables could influence one another. First, we applied multivariate linear regressions to estimate the independent contribution of each predictor to downstream variables within this hierarchical framework. Only statistically supported paths ($p < 0.05$) were retained in the final causal diagram, and standardized beta coefficients from the regression models were used to represent path strengths. The following model was applied:

$$SSS \sim Dia + Chest + Lung + Inhale$$

$$HR \sim Dia + Chest + Lung + SSS + Inhale$$

$$MaxFlow \sim HR + SSS + MeanPV$$

$$MeanPV \sim MaxFlow + HR + SSS$$

$$Dis \sim MaxFlow + MeanPV$$

$$NET \sim Dis + MeanPV + MaxFlow$$

As a second step, SEM (lavaan package in R version 4.4.3) was used to evaluate the directionality of statistically significant relationships identified in the multivariate analysis. Respiratory functions (lung area displacement, inhale time, chest displacement, and diaphragm displacement) were consistently specified as initial predictors, and CSF displacement and CSF net flow were defined as the final outcomes.

Because this framework is regression-based, it does not yield conventional global SEM fit indices such as the comparative fit index (CFI), root mean square error of approximation (RMSEA), Tucker–Lewis index (TLI), and standardized root mean square residual (SRMR), which are defined only for a single joint covariance-based SEM. To provide a complementary evaluation, a separate exploratory acyclic SEM was performed and reported only as fit index values in the Results. To ensure model identifiability and numerical stability, the reciprocal paths between MeanPV and MaxFlow were replaced with a residual covariance (*MeanPV ~ MaxFlow*), and covariances were specified among respiratory predictors (Dia, Chest, Lung, Inhale) to account for shared variance.

### Statistical analysis

All statistical analyses were performed using GraphPad Prism (version 10.4.1, GraphPad Software) and R (version 4.4.2). For group comparisons, Student's t-tests, and paired two-tailed t-tests were applied for within-subject comparisons using the built-in statistical algorithms in GraphPad Prism. Correlation analyses were conducted separately within the T and NT groups, and linear regression trends were estimated using the same software. Pearson's correlation coefficient matrices were computed in R to examine associations between CSF flow parameters, physiological measures, and respiratory features.

### Phantom analysis

For phantom data, nonparametric Kruskal–Wallis tests were used to evaluate differences across three cardiac pulsation conditions and three respiratory modulation conditions for mean velocity, displacement, and net flow. When significant, Dunn's post hoc test was applied to determine pairwise effects.

### Confidence interval (CI) analysis

CI analyses were performed to compare CSF flow metrics across breathing conditions, using NT-RB as the reference group. Mean ± 95% CI values were calculated in R to assess statistical trends in NT-DB, T-RB, and T-DB.

### Respiratory rate regression analysis

To account for potential confounding effects of RR, an additional analysis was performed in which RR was regressed out from the CSF flow metrics, mean speed, displacement, and net flow, within the FM and LV. Specifically, each metric was modeled as a function of RR (e.g., *FM_MeanSpeed ~ RR*), and the resulting residuals ($\varepsilon$) representing RR-independent variance were extracted (*FM_MeanSpeed ~ RR + $\varepsilon$*, etc.). Group comparisons between T and NT participants were then repeated using these residual values to confirm that observed differences in CSF dynamics were not driven by variations in RR. Results from this analysis are presented in Supplementary Fig. S6.

### Comparison between upper and lower net flow levels

To examine respiratory and physiological characteristics associated with higher CSF transport, participants were divided into two groups based on the median value of FM net flow, representing the upper and lower 50% of the distribution. Comparisons between these groups were performed for respiratory, physiological, and CSF flow features in both the FM and LV. This analysis was used to identify features linked to greater CSF net flow, independent of training status. Results are shown in Fig. 4.

**Reporting summary**

Further information on research design is available in the Nature Portfolio Reporting Summary linked to this article.

## Data availability

The processed data used to generate all figures are provided in the 'Source Data' file included with this paper. Raw MRI and physiological datasets are available under restricted access due to IRB and privacy regulations. De-identified data can be requested from the corresponding author, Paul H. Min (min.paul@mayo.edu), for academic research use and will require a data use agreement. Requests will receive a response within 10 business days and, if approved, data will be shared via secure transfer. Data will remain available for at least 5 years after publication. Source data are provided with this paper.

## Code availability

Custom analysis code used in this study is available from the first author, Seokbeen Lim (lim.seokbeen@mayo.edu), for academic research use. Requests will receive a response within 10 business days. No external software beyond standard analysis tools was used.

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

## Acknowledgements

This study was supported by the Linse Bock Foundation (Min and Lapid), NIH R37 AG011378 (Jack), and NIH C06 RR018898 for the Mayo Clinic Opus Advanced MRI Facility. We are deeply grateful to the study participants for their time and commitment. We also thank the Seokmun Domun, a nonprofit organization, for their support in participant recruitment and guidance on breathing practice. Special thanks to our study coordinators, Sue Walsh and Tim Waters, for their contributions to participant management and data collection. Additional thanks to Corey Woxland for technical support and assistance with MRI operations.

## Author contributions

S.Lim, P.M.C., Z.M., S.Lee, J.G., J.G.R., E.E.B., J.H., C.R.J., R.C.P., M.I.L., V.J.L., D.J. and P.H.M. conceived and designed the study. S.Lim, Y.C., D.C.K., P.K., S.Lee, S.G., I.T.M., D.K., M.H.I. and J.G. developed and implemented the methodology. MRI data acquisition and investigation were performed by S.Lim, D.N.J., M.K. and M.N. The flow phantom study was performed and analyzed by S.Lim, D.N.J., P.J.R. and M.H.I. S.Lim, P.M.C., Y.C., M.N., P.K., D.J. and P.H.M. prepared the visualizations. W.K.K. supervised the statistical analysis. Funding was acquired by P.M.C., J.H., C.R.J., R.C.P., M.I.L., V.J.L., D.J. and P.H.M. S.Lim, P.M.C., D.N.J., M.K., Y.C., M.N., D.C.K., P.K., Z.M., J.G.P., D.K., M.H.I, J.G., J.G.R., E.E.B., J.H., C.R.J., R.C.P., M.I.L., V.J.L., D.J. and P.H.M. wrote the original draft of the manuscript. All authors contributed to reviewing and editing the manuscript under the supervision of P.M.C., V.J.L., D.J. and P.H.M.

S.Lim, P.M.C., Z.M., J.G.P., J.G.R., E.E.B., J.H., C.R.J., R.C.P., M.I.L., V.J.L., D.J. and P.H.M. contributed to data interpretation and discussion of results throughout the study. P.M.C. and P.H.M. supervised and directed the project. All authors contributed to the final manuscript.

## Competing interests

The authors declare no competing interests.

## Additional information

Seokbeen Lim[1], Petrice M. Cogswell [1] ✉, David N. Jacobson[1], Mahathi Kandimalla[1], Yurim Choi [1], Marin Nycklemoe[1], Daniel C. Kim [1], Pragalv Karki [1], Phillip J. Rossman[1], Zona McKenzie[1], John G. Park [2], Sangwon Lee[3], Sandeep Ganji[1,4], Walter K. Kremers[5], Ian T. Mark [1], Daehun Kang [1], Myung-Ho In[1], Jeffrey Gunter[1], Jonathan Graff-Radford[6], Eduardo E. Benarroch[6], John Huston III[1], Clifford R. Jack Jr. [1], Ronald C. Petersen[6], Maria I. Lapid[7], Val J. Lowe[1], Daehyun Jung[3] & Paul H. Min [1,8] ✉

[1]Department of Radiology, Mayo Clinic, Rochester, MN, USA. [2]Department of Pulmonary and Critical Care Medicine, Mayo Clinic, Rochester, MN, USA. [3]Seokmun Domun, Suwon, Republic of Korea. [4]Philips Healthcare, MR R&D, Rochester, MN, USA. [5]Quantitative Health Sciences, Mayo Clinic, Rochester, MN, USA. [6]Department of Neurology, Mayo Clinic, Rochester, MN, USA. [7]Department of Psychiatry and Psychology, Mayo Clinic, Rochester, MN, USA. [8]Department of Physiology and Biomedical Engineering, Mayo Clinic, Rochester, MN, USA. ✉e-mail: cogswell.petrice@mayo.edu; min.paul@mayo.edu

