## [Transparent Peer Review file · Nature Communications]

Human Cerebrospinal Fluid Net Flow Enhanced by Respiration During the Awake State

Corresponding Author: Dr Paul Min

Version 0:

Reviewer comments:

Reviewer #1

(Remarks to the Author)

Summary / Innovation / Significance

Understanding mechanisms driving and/or altering CSF dynamics is critical to better understand its clinical impact, and implications on CNS diseases as CSF is crucial for CNS health. Authors in this study have developed a methodology utilizing velocity encoding MRI for CSF measurements and diaphragm MRI for modeling diaphragm motion, and examined primarily respiratory-driven CSF dynamics in a set of participants (breathing trained, non-trained, and people with Alzheimer's disease). Study results suggest breathing may serve as a potential intervention for brain health via improvement CSF dynamics.

This is such an important study - highlighting the importance of breathing for CSF dynamics, and potentially as part of complementary and integrative medicine modalities to support brain health. However, it is not ready for a publication.

Critiques/Comments

I was pleased to review this paper until I started to read Main section and visited to Materials & Methods (M&M) section many times to understand what the actual study population was (which is not indicated in Abstract by the way), and how grouping was done, who was trained vs non-trained; all of which, unfortunately, was a disappointing experience. I started to make comments for authors to revise, but before providing more, I believe it is fair if authors revise the entire paper for a smooth and cohesive study flow with clear information of study population, measurements, and results, all of which will improve readability, and will make it easier to review.

Major

Please revise the manuscript appropriately for a cohesive and smooth flow, providing simple flow of study experimental tasks, including inclusion/exclusion criteria, study flow chart (similar to consort diagram) to begin with.

Feedback for revisions:

Abstract: Abstract shall provide information on methods including subject number, mean (SD) age, sex, number of groups (healthy: trained, non-trained, and AD), and statistically significant results of the comparisons. It takes couple of reads to understand what the study population entails to.

Provide CSF measurement locations in the Abstract.

Main: Last paragraph indicates methodology was examined in two groups (trained, T and non-trained group, NT). Yet, "Group-level comparison of baseline CSF dynamics" begin with three groups (trained, healthy control, and AD)..Please clarify the study populations clearly in the Main last paragraph, and abstract.

Materials & Methods:

1. Please make it is easy for readers to understand the study population groups without reading several times. Study seems to include healthy participants which is further divided into two groups: trained (N=17) and non-trained (N=21), and also includes people with AD (N=10). But then following sections on M & M indicate including T=17 with 11 healthy controls and 10 AD. And Table S1 Demographics show T (17), HC (32), and AD (10). Total =59. And there is the HC=32 (NT + AD???)?

Please determine the right grouping, use associated acronym, either T, HC, and AD, or T, NT and AD, or HC (T, NT) and AD, which all depends on what healthy control is: whether healthy control vs AD or for T vs NT comparison. Then please clarify each group's N# and in Table S1, apply modifications, and include mean (SD) age in years, age range in years, sex (provide also %) for each and every group after determining the right grouping. What's the p-value in Table S1 for? Please clarify.

2. After determining right grouping, please provide inclusion/exclusion criteria for each group including AD group. Provide very brief information from previous study regarding recruitment of HC and AD, and what AD population neuropsychological exam entails without the need to refer to the previous study, ideally provide a Table S. Since this study includes data from a previous study, please make it more clear in Participants section to determine which subject data is from the current study, and which from a previous study, and what exclusion and inclusion criteria for previous and new study were.

3. "To ensure consistency, we selected a training method that focused explicitly on breathing practice and included only those who had engaged in a single, continuous program over an extended period."

Please explain what it means, and how was the inclusion criteria were determined? Participants were recruited based on what frequency/duration of their practice? e.g., 3 times/week 20 minutes, etc.? And with at least 1 year of practice? Practitioners with 1 year vs 30 years will have a wide range of variability inn their results. How was this range clarified when analyzing data?

4. "Participants in the NT group were screened using a questionnaire to confirm no prior sustained formal training in breathing practices (more than six months) such as diaphragmatic breathing, lower belly breathing, yoga, or mindfulness."

How was this determined in terms of cut-off time close to the study? Was there a defined cut-off time, e.g., if someone has done 5 months breathing practice right prior to the beginning of the study, would they be included? In other words, if someone has concluded an 8-week MBSR prior to the study, would they be excluded? Please clarify inclusion/exclusion criteria as both the definition of long-term and duration of practice up until the start of the study even if it was 5 months (<6) is critical for complementary and integrative medicine modalities and their influence on study outcomes and/or clinically. Considering many clinical trials include 8 to 12-week interventions to understand pre- and post-intervention outcome measures, a potential participant practicing for instance 3 month right until the start of the study would be excluded.

5. "The practice combines physical movement with focused attention on a specific point in the lower abdomen to encourage natural, sustained breathing habits."

Please define physical movement whether this is movement of abdomen alone or actual synchronized physical motions along with breathing like in Tai-Chi, Qi-Gong, etc.

6. For a smooth read: Please remove CSF measurement related sections within Participants section and move to a new CSF measurements subheading or move to MRI Acquisition sections. It is confusing to read the paper with its current form. Change Breathing Protocol to MRI Breathing Protocol to differentiate Breathing Training vs Breathing practices performed during MRI acquisitions. Overall, please revise the experimental methodology sections to ensure a smooth and cohesive flow.

7. Include a study flow chart (e.g., similar to consort diagram, including number and reasons of excluded participants from each group).

Reviewer #2

(Remarks to the Author)

To the authors:

This study provides a meaningful and timely contribution to the understanding of cerebrospinal fluid (CSF) dynamics, particularly by highlighting the potential impact of breathing methods on CSF inflow via autonomic modulation of the cardiovascular system. The application of multiple imaging techniques to explore CSF dynamics at the foramen magnum and lateral ventricles is commendable, and the focus on diaphragmatic movement introduces an important physiological perspective that broadens the scope of CSF research. However, the classification of experimental groups according to breathing practices—arguably the core variable of this study—would benefit from further clarification and scientific rigor. The current design limits the interpretability of key findings. The reviewer suggests the authors consider the following points to strengthen the scientific validity and overall impact of their work:

Major Comments:

1. Group classification and Variable Control: The current categorization of the "Training" group based on a single breathing technique (Seokmun breathing) lacks a precise scientific definition and variable control. For a more rigorous approach, we

recommend identifying and measuring key physiological parameters that differ between trained and control subjects (e.g., respiratory rate, tidal volume, I:E ratio). Analyzing how each of these factors independently and interactively affects CSF dynamics would improve mechanistic understanding. For example, a quantitative correlation between diaphragmatic movement and other physiological variables could clarify causal pathways.

2. **Autonomic and Cardiovascular Evidence:** While the hypothesis regarding autonomic modulation of the cardiovascular system is conceptually appealing, direct evidence—such as group-wise differences in HR, BP, and autonomic markers (e.g., catecholamines, cortisol)—was not presented. Future studies could strengthen this argument by evaluating whether similar CSF changes are observed in other autonomic conditions (e.g., during exercise or sleep) to validate the generalizability of the proposed mechanism.

3. **Diaphragmatic Movement vs. CSF Dynamics:** Although diaphragmatic movement is discussed as a critical factor, some data suggest notable CSF differences despite minimal differences in diaphragm displacement between NT and T groups. Addressing this apparent discrepancy would provide greater insight into the underlying physiological mechanisms.

4. **Baseline Physiological Metrics:** Given that trained individuals exhibited higher mean absolute CSF velocities at rest, reporting baseline cardiovascular and respiratory variables (e.g., HR, RR) is essential to rule out pre-existing physiological differences as confounding factors.

5. **Intra-individual Breathing Comparison:** A direct comparison of resting vs. deep breathing within the same individuals—particularly in both trained and untrained groups—would greatly enhance the interpretability of the training effect and help isolate the influence of breathing depth.

6. **Correlation Analysis Between Groups:** To support the claim of a positive correlation between HR displacement and lateral ventricle (LV) CSF displacement, correlation analyses should be performed separately for the trained and untrained groups, ensuring group-specific effects are not conflated.

Minor Comments:

1. Use "mean speed" instead of "mean velocity" when referring to the absolute value of velocity.
2. Lines 166–170 are speculative and not directly supported by the presented data; there is no clear link between CSF inflow changes and glymphatic function.
3. The statistical analysis section requires more detail, including complete p-values in all figure panels. For example, Figures 3f and 3g lack p-values.
4. Figure 2: To improve clarity, present correlation statistics separately for the T and NT groups.
5. Please verify the P-value in Table S1—is 0.000 statistically appropriate?
6. Re-check the P-value in Supplementary Figure 2f. The legend states a significant difference, yet the P-value appears to be 0.512.
7. Include individual data points (dots) on all bar graphs to enhance data transparency.
8. In figures such as S6b/d and S7b/d, redraw the y-axis starting from 0, or use break symbols if a truncated axis is needed to show variation clearly.
9. Indicate the number of individuals included in Fig S6b in the legend.
10. Figure 3a is not referenced in the manuscript.
11. Justify the extraction of voxels related to cardiac signals during lateral ventricle analysis.
12. Line 200: Please confirm whether "LV displacement" is the correct term.
13. Line 203–204 and Figure 3e: Clarify which group the trained individuals were compared to and how the stated P-value was derived.
14. Add P-values to Figures S9a and S10a to improve readability.
15. Clarify the CSF net flow patterns at the foramen magnum and lateral ventricle by adding sub-headers or detailed labels in relevant graphs.
16. The order of supplementary figures does not follow the manuscript's flow (e.g., Supplementary Figure 11); please revise accordingly.
17. The hypothesized role of the superior sagittal sinus (SSS) as a driver of CSF flow needs mechanistic support. Consider including velocity-encoded MRI data, particularly after Q-collar application, to support this claim.
18. Provide the P-value for Figure 5b.
19. Figure 6 appears to present new data but is only discussed in the Discussion section. Consider relocating this to the Results section for coherence.

Reviewer #3

(Remarks to the Author)

In the manuscript by Lim et al, the authors propose that controlled breathing exercises can enhance CSF fluid flow within the lateral ventricles. The authors use MRI techniques on populations of trained and untrained subjects during breathing, in an attempt to measure differences in CSF flow between these groups. The authors proposed that breathing exercises may be a lifestyle intervention to improve CSF circulation and potentially waste clearance. While this is an attractive and novel hypothesis, the data in the current manuscript does not support this concept.

Major comments

1. The title does not reflect the content. The most pronounced effects of respiration on CSF dynamics occur at the foramen magnum rather than the lateral ventricle. There are no meaningful significant effects on CSF displacement or flow between non-trained and trained individuals in Figure 3b or e. I also fail to see a significant effect of deep breathing on these parameters compared to regular breathing in either group.
2. The authors appear to have reused previous data in the study. The paper integrates 11 healthy controls from a prior published study and rebrands them as part of a healthy control (HC) group with the new NT cohort stating, "we combined the 11 previously studied healthy participants with the 21 NT participants... and labelled this combined group as the healthy control (HC) group." This pooled group lacks internal consistency, as the reused participants did not undergo the same protocols (e.g., deep breathing or diaphragm MRI), yet are directly compared. This practice inflates sample size and may confound interpretations of CSF flow baselines. The inclusion of Alzheimer's disease patients from this earlier study is also problematic. There is no reasoning given why these patients are included in the current study. The data from these patients is included in Fig 1 and Fig 4, but not in other figures.
3. The manuscript does not clearly separate the methods, results, and discussion sections, often mixing explanations of the analysis with descriptive text. For example, references to "Fig. 4" and "the final causal diagram" appear in places where it is unclear whether the authors are describing their procedures or interpreting the findings. Some figures are introduced in the discussion, such as Figure 6. Also, the important flow phantom validation, which should be described in the methods, is only mentioned late in the discussion. Supplementary Figure S16 is never referenced or explained in the main text. Together, this disorganisation reduces the clarity of the paper, making it difficult to follow the study's narrative.

Version 1:

Reviewer comments:

Reviewer #1

(Remarks to the Author)

Summary

As indicated in initial review, the central question of this manuscript is timely and important, with strong implications for advancing integrative health research within the context of CNS health. The findings are potentially impactful and of broad interest. The authors have made significant progress in addressing previous review comments. However, further revisions are needed to strengthen the manuscript. Please see the critiques and comments below (rather long), that are intended to improve the clarity, rigor, and accessibility of the work, all to ensure study contributions are communicated in the strongest possible way.

Major Comments

Methods/Study Flow: The overall experimental approach and methodology are appropriate (except see below for a major concern). The description of methods, however, could be improved for readability via presenting them in a more chronological order that follows the actual study workflow. This would help readers better follow the logic of study design and data collection and remove remaining confusion from initial manuscript.

Choice of CSF Metrics: The authors report "mean CSF speed" as a primary outcome, rather than peak, minimum, or peak-to-peak CSF velocity values. This choice raises concerns about interpretability. Because CSF dynamics are characterized primarily by pulsatility and oscillations, mean values may obscure physiologically meaningful fluctuations.

- The authors should clarify why mean CSF speed was prioritized. In particular, why was speed (magnitude only) chosen instead of velocity, which preserves directionality and is more physiologically informative?

- Since the study investigates the effects of regular versus deep breathing on CSF dynamics, peak (maximum, minimum, and/or peak-to-peak) velocity measures are more valuable for quantifying the true impact of breathing on pulsatile CSF. Including such measures would also allow clearer interpretation of differences across breathing patterns (regular vs. deep) and participant groups (T vs. NT), and would align better with the physiological rationale while capturing true dynamics (e.g., modulation of CSF by diaphragm motion).

The choice of additional CSF metrics -esp. in correlation analysis- appears somewhat selective, and the rationale for their inclusion vs. other commonly used measures is not fully clear.

Lastly, as a general note, there should be a greater clarity and consistency in the terminology used for the CSF measure. What PC-MRI measures is the velocity of the CSF pulsatile motion/movement, and secondary metrics like CSF net flow and displacement are derived from it. Yet often studies may use the term CSF flow or circulation. While authors may have their own framing, it would be important to adopt a consistent use of a term: CSF dynamics or pulsatile CSF motion or oscillatory movement, which will strengthen the manuscript, and will also help avoid confusion for readers.

Other comments:

Abstract:

1. Please use "mean (\pm SD)" in age related sections, e.g.: mean (\pm SD) age 58.1 \pm 17.3 years.

2. "All underwent real-time velocity-encoding magnetic resonance imaging (MRI) to assess CSF flow at the foramen magnum (FM) and lateral ventricle (LV)."

Please add "during regular and deep breathing" to provide methodological information upfront.

3. "Our findings identify respiration in the awake state as a modifiable, noninvasive mechanism that influences involuntary functions such as CSF circulation."

Authors in this study compute "CSF speed, net flow and displacement". It is important to note that displacement reflects oscillatory and bidirectional motion rather than unidirectional transport. Similarly, net flow reflects directional bias in CSF movement, may be small, and not sustained enough to represent circulation. CSF circulation on the other hand involves systematic / global turnover through ventricular system and SAS, and requires turnover pathways rather than local oscillatory motion dynamics. Thus, the use of "CSF circulation" in above statement is not supported by the findings.

4. "This suggests that breathing may serve as a viable lifestyle intervention to support brain health by improving CSF-mediated homeostatic mechanisms, with potential future clinical applications in dementia prevention and care."

While breathing is known to regulate CSF dynamics, mechanistic pathways between brain health and respiratory driven CSF-mediated homeostatic processes remain unclear. Further, current research does not establish how such mechanisms could be linked to dementia prevention, and authors are not examining CSF-ISF driven glymphatic clearance but rather investigating the impact of breathing in long-term breathing practitioners vs. novices. Thus, while the suggestion above may be plausible, it is not sufficiently supported by the evidence presented. It can be, and in my opinion, certainly shall be included in discussion.

5. Please also include in the results: how regular vs. deep breathing findings differ among practitioner vs. novice participants.

Introduction:

1. References included in introduction can be reviewed for a revision as some of the citations may not be representing the statements, while some statements do not include the full list of citations included in references.

2. Similarly, as in abstract, please consider modifying

"All participants underwent CSF flow measurement in the brainstem and the lateral ventricles using velocity-encoding MRI during both resting wakefulness and voluntary slow, deep breathing." as

"All participants underwent CSF flow measurement in the brainstem and the lateral ventricles using velocity-encoding MRI during regular breathing and voluntary slow, deep breathing, both acquired during wakefulness" or something on that order to mention regular and deep breathing terminology upfront.

3. In last paragraph, please consider including a line for phantom measurements for validation of the MRI technique prior to indication of examining two groups.

Results:

1. Figure S1. Inclusion of questionnaire in study flow chart seems wrong. Perhaps it was meant to be included prior to enrollment/consent process since those within the NT group would be determined prior to enrollment based on questionnaire.

2. p values throughout the paper shall be included as $p=X$ or $p<X$ rather than $p = X$, or $p < X$.

3. Use of acronyms for the rest of the manuscript after it was defined for consistency, such as HR for heart rate, RB for regular breathing, DB for deep breathing, etc... once it was defined.

4. Please clarify the use of speed instead of velocity in human experiments, while the term velocity is used for phantom experiments. And as in also noted above & in Methods section critiques, please clarify use of mean CSF speed instead of max and min, or peak-to-peak CSF velocity for a study quantifying the impact of breathing during distinct breathing patterns, to capture dynamic ranges.

5. While the T group shows higher mean CSF speed values compared to the NT group under both RB and DB, within-group comparisons are surprisingly similar. For example, at the FM level, T participants exhibit 0.83 ± 0.05 vs. 0.84 ± 0.06 (RB vs. DB), and NT participants show 0.66 ± 0.03 vs. 0.66 ± 0.03 . These results contrast with prior studies reporting a clear increase in CSF velocity during deep or forced breathing compared to natural/regular breathing.

Looking at a representative participant in Figure S12, the flow rate amplitude appears similar between RB and DB, although the phase is clearly different. Moreover, the DB condition seems more reflective of slow breathing rather than deep breathing, which may explain the absence of increased amplitudes. Or, perhaps these results from the use of mean speed compared to max/min or peak-to-peak CSF velocity, which as indicated previously raises a major concern for the methods, thus findings.

6. "In this study participants in the T group performed the 'breathing practice' component of these sessions."

7. Does it mean only the T participants performed the deep breathing based on training style, but the NT group performed a

different deep breathing pattern? Or both groups performed same instructions? Please clarify.

8. Please provide phantom study section prior to human subject research methodology for the validation of the technique, and readability of the study flow.

9. Fig 2d Condition 6: Is the overlapping phase signature (unlike in Conditions 4-5)- due to reversed signals during two measurements?

10. Please provide Figures and Supplementary figures in the order mentioned in the manuscript.

11. Flow analysis (for the CSF ROI selection and extraction of CSF velocity time series): The manuscript currently lacks clarity regarding the process for selecting the CSF ROI and the CSF signal components, as well as the bandwidths applied. Specifically:

- It is stated that a bandwidth of [0.8–2 Hz] was used primarily for cardiac-driven CSF ROI selection, and [0.01–0.5 Hz] for the respiratory band. However, this does not explain what bandwidth was ultimately used for computation of the CSF velocity time series, and the derived measure of mean CSF speed, and derived cardiac- and respiratory CSF components. This raises a major methodological concern about whether different voxels were used for cardiac- vs. respiratory-driven CSF signals. If so, please justify why.

Please clarify explicitly:

1. Was CSF signal ROI selection different for cardiac and respiratory components? If yes, why?

2. What bandwidth was used for the computation of CSF velocity – to then derive mean CSF speed, which is reported as one primary outcome.

3. What bandwidth was used for the cardiac component of CSF signals, and what bandwidth was used for the respiratory component? Not just for voxel selection but for the actual analysis?

- The methodological choice of using CSF speed rather than CSF velocity also requires clarification as indicated before. RT-PCMRI provides velocity which preserves directionality, whereas speed removes this directional information and reduces interpretability. In phantom flow experiments, velocity is reported, but in the human experiments the manuscript uses the term speed. Please clarify this discrepancy.

- Finally, since the study investigates two distinct breathing types (RB and DB) and diaphragm movement, it would be more appropriate to quantify maximum CSF (systolic), minimum CSF (diastolic), and/or peak-to-peak CSF velocities rather than mean speed. Averaging CSF speed obscures and likely underestimates the true magnitude of the instantaneous maxima and minima (as in Fig S14a CSF velocity time series and red line showing mean CSF speed) during RB vs DB, which then likely underestimates the true impact of breathing on CSF dynamics during two breathing patterns. Since this study relies on comparing RB & DB among T & NT groups, it is critical to compare actual dynamic range of CSF.

- Lastly, this methodological choice may explain why mean CSF speed appears very similar between RB and DB conditions, as indicated in one of my comments above-. And, I suspect/ suggest the reported “mean CSF speed” may reflect the cardiac-driven component of CSF motion rather than instantaneous CSF, which does not differ substantially between breathing patterns in this study.

12. Request for time- and frequency-domain representation of instantaneous CSF velocities: To aid methodology and interpretability, I strongly recommend that the authors provide representative examples of instantaneous CSF velocity time series and corresponding frequency domain signals. Specifically, please include:

-Time- and frequency-domain plots of band-pass filtered CSF velocities within [0–2 Hz] if that's what used for CSF speed computations, for one participant from each group/condition (T-RB, T-DB, NT-RB, NT-DB).

- Time-domain velocity traces should be presented as in Fig. S14, with the instantaneous CSF velocity time series and the mean CSF speed indicated by a red line.

-The corresponding frequency-domain plots should be shown alongside, highlighting the dominant respiratory and cardiac peaks.

A similar figure is already included (Fig. S12), but it shows flow rates rather than velocity time series. Providing the velocity-based plots will substantially improve clarity, please demonstrate how filtering was applied, and allow readers to better understand how the reported outcomes were derived, which will remove the methodological concerns regarding the interpretability and discussion of findings.

13. An expert opinion on diaphragm motion analysis may be needed.

14. Please clarify the selection/inclusion of CSF metrics for correlation.

15. T group practitioners (including authors included) who are aware of the study aims may intentionally or unintentionally exaggerate or modify their breathing during scanning, potentially amplifying apparent group differences.

16. And more broadly, as with many MRI-based physiological studies, it is not entirely clear whether breathing patterns in the MRI match the practitioners true practice patterns. These should be acknowledged in the Discussion, ideally within a defined Limitations section, for transparency.

Reviewer #2

(Remarks to the Author)

The authors adequately addressed my comments. Congratulations!

Reviewer #3

(Remarks to the Author)

Overview

The manuscript now presents a more refined study integrating phantom validation, physiological recordings, and MRI analysis to examine how breathing practices influence CSF dynamics. The Results are more clearly structured, with improved consistency and better integration of supplementary figures into the main text. The definitions of CSF flow features and the use of SEM with correlation analyses strengthen the mechanistic interpretation, and the phantom validation remains a key strength. The revisions address earlier concerns around data reuse and organization, which improves readability and focus. Some questions remain, particularly regarding how robustness of the SEM is reported and how multiple comparisons are handled, but the clarifications made so far represent clear progress.

Major comments

1. In group comparisons (Fig. 3, S11), trained participants show longer inhale length, greater diaphragm displacement, and increased HR and SSS displacement during regular breathing. A potential concern is that the significantly lower respiratory rate in the trained group could partly account for these differences. The authors note that both HR and respiratory rate were incorporated into the SEM and multivariate models as mechanistic features rather than uncontrolled confounders, which is a reasonable approach. To strengthen confidence in this interpretation, it may be helpful to make this distinction clearer in the text and, if possible, to include a simpler sensitivity analysis (for example, adjusting group comparisons for respiratory rate) to demonstrate that the findings are robust across different analytical frameworks.
2. The tertile-based analysis (Fig. 4, S13) is a creative way of moving beyond training status and looking for generalisable features, but tertiles may not be the most stable approach given the modest sample size. It might be worth clarifying whether alternative splits (e.g., quartiles or median) produce comparable results. More broadly, the statistical handling of multiple comparisons remains a little unclear. The study reports many tests (t-tests, paired comparisons, regressions, correlations, and Kruskal–Wallis post hoc analyses) but no correction method (such as false discovery rate or Bonferroni) is described. Without adjustment, the risk of false positives is increased significantly. In addition, although the Methods indicate that SEM model fit was assessed with indices like RMSEA, CFI, TLI, and SRMR, these values are not reported in the Results. Including the fit indices would make it much easier for readers to judge the adequacy of the causal framework.
3. The observation of a small but positive mean net CSF flow at the foramen magnum (~0.05 ml; Figs. S13–S14) is intriguing but raises interpretive challenges. Taken at face value, this could suggest net upward CSF transport from the spinal canal, which would have important implications given ongoing debate about CSF production sites. At the same time, the Methods indicate that factors such as ROI placement, baseline drift, and integration of oscillatory signals could introduce subtle offsets, and while phantom validation shows good reproducibility in controlled conditions, it cannot fully rule out small in vivo artifacts. For this reason, it would be valuable for the manuscript to discuss this finding explicitly, considering both the possible physiological interpretation and the methodological caveats, and to situate it within the broader literature so that readers do not over-interpret the result.

Minor comments

1. Figure S11 highlights group differences in chest (thoracic) versus diaphragm displacement during deep breathing, but these are only briefly mentioned in the text. A short explanation linking these patterns to different breathing strategies would help clarify their relevance.
2. The representative case in Figure S12 includes only a trained participant. Here, adding a contrasting non-trained example would make the claim of altered coupling with practice more compelling.
3. Several supplementary figures, particularly S11 and S13, contain results central to interpretation yet are only briefly referenced. The authors should weave these figures more directly into the main text which would reduce the need for readers to flip back and forth.
4. The phantom validation would benefit from separating cardiac-like and respiratory-like oscillations into distinct subsections for clarity.
5. The supplementary videos appear to quantify respiratory area using straight-line approximations, which cut-out the natural curvature of the breathing trace. While the Methods make clear that quantitative diaphragm motion analysis was performed with DeepLabCut, it remains ambiguous whether the simplified videos were used quantitatively or merely for illustration. This distinction should be clarified, as linear approximations could underestimate displacement.
6. The link to glymphatic function in the Discussion is interesting, though since glymphatic clearance (along perivascular spaces) was not directly measured, it may help to phrase this connection a bit more cautiously. Clarifying that the present findings reflect macroscopic CSF dynamics, which may provide context for but are distinct from perivascular clearance processes, would strengthen the interpretation without overextending the scope.

Minor errors

1. A few errors should be corrected to improve accuracy and readability. For example, in the section on baseline physiology, the heart rate for deep breathing in the trained group is reported as “66.0 ± 0.16 bpm.” “LV CS displacement” is written instead of “LV CSF displacement.” The formatting of CSF-related acronyms varies across sections.

Version 2:

Reviewer comments:

Reviewer #1

(Remarks to the Author)

The authors have adequately addressed the previous review comments. The revisions improved the clarity and quality of the manuscript. I have no additional comments at this time unless other reviewers have, and support moving forward with publication. Congratulation to the entire study team.

Reviewer #3

(Remarks to the Author)

The manuscript is much improved and the comments have been mostly addressed.

We had one remaining point in regards to the SEM model fit as part of the original major comment 2:

Including the exploratory joint SEMs and their fit indices is appreciated and adds transparency. That said, the reported values (e.g., CFI below 0.9, RMSEA above 0.1) suggest that the overall model fit was not very strong. While the explanation about collinearity and model restrictions is reasonable, it would be good to acknowledge this more directly so that readers understand the limitation.

A simple sentence noting that the joint SEMs showed poorer global fit, and that this limits the strength of causal interpretation even though the directional trends were consistent, would address this clearly.

❖ **Reviewer #1 (Remarks to the Author):**

● **Summary / Innovation / Significance**

Understanding mechanisms driving and/or altering CSF dynamics is critical to better understand its clinical impact, and implications on CNS diseases as CSF is crucial for CNS health. Authors in this study have developed a methodology utilizing velocity encoding MRI for CSF measurements and diaphragm MRI for modeling diaphragm motion, and examined primarily respiratory-driven CSF dynamics in a set of participants (breathing trained, non-trained, and people with Alzheimer's disease). Study results suggest breathing may serve as a potential intervention for brain health via improvement CSF dynamics.

This is such an important study - highlighting the importance of breathing for CSF dynamics, and potentially as part of complementary and integrative medicine modalities to support brain health. However, it is not ready for a publication.

● **Critiques/Comments**

I was pleased to review this paper until I started to read Main section and visited to Materials & Methods (M&M) section many times to understand what the actual study population was (which is not indicated in Abstract by the way), and how grouping was done, who was trained vs non-trained; all of which, unfortunately, was a disappointing experience. I started to make comments for authors to revise, but before providing more, I believe it is fair if authors revise the entire paper for a smooth and cohesive study flow with clear information of study population, measurements, and results, all of which will improve readability, and will make it easier to review.

=> Thank you for your encouraging notes and helpful guidance. We have reorganized the manuscript by carefully integrating the reviewers' comments. The main change is that we focused this paper on the trained (T) vs. non-trained (NT) group study. We excluded all data from the previous publication, which included both AD participants and some healthy controls. To strengthen the comparison, we added seven new participants (n = 3 T; n = 4 NT), resulting in final trained (n = 20) and non-trained (n = 25) groups. We also revised the Abstract, clarified the study populations and inclusion/exclusion criteria, restructured the Methods for a smoother flow, and added a study flow chart. We hope these revisions address the reviewers' concerns and improve the clarity and readability of the manuscript.

• **Major**

Please revise the manuscript appropriately for a cohesive and smooth flow, providing simple flow of study experimental tasks, including inclusion/exclusion criteria, study flow chart (similar to consort diagram) to begin with.

=> We thank the reviewer for this helpful comment. We have revised the manuscript to improve clarity and flow, and we have added a study flow chart (Supplementary Figure S1) summarizing the study design, enrollment process, and inclusion/exclusion criteria. In addition, we have updated the *Materials and Methods, Participants* section to clearly describe the study population and criteria.

Figure S1. Study flow chart. Flow diagram showing participant screening, exclusions, group allocation (trained vs. non-trained), and final numbers included in each imaging modality (FM-PC-MRI, LV-PC-MRI, SSS-PC-MRI, diaphragm MRI) and corresponding analysis sets.

- **Feedback for revisions:**

Abstract: Abstract shall provide information on methods including subject number, mean (SD) age, sex, number of groups (healthy: trained, non-trained, and AD), and statistically significant results of the comparisons. It takes couple of reads to understand what the study population entails to.

Provide CSF measurement locations in the Abstract.

=> Thank you for this important suggestion. We have revised the Abstract to clearly present the study demographics, methods, and statistically significant results. CSF measurement locations have also been explicitly included in the Abstract.

(Revised Abstract)

“Abstract

Cerebrospinal fluid (CSF) circulation plays a crucial role in maintaining brain homeostasis by delivering nutrients, transmitting immune signals, and clearing waste products. While cardiac activity primarily drives the pulsatile flow of CSF, respiration has been shown to facilitate low-frequency oscillations and contribute to bulk flow. Recent studies suggest that enhancing respiratory function may be an effective intervention to modulate CSF dynamics. This study included 20 individuals with long-term formal training in Seokmun Hoheup (mean age 58.1 ± 17.3 years; 8 females) and 25 controls with no formal long-term breathing practice (mean age 49.2 ± 20.2 years; 12 females). All underwent real-time velocity-encoding magnetic resonance imaging (MRI) to assess CSF flow at the foramen magnum (FM) and lateral ventricle (LV). Seokmun Hoheup trained participants showed distinct CSF flow differences compared to controls at both the FM and LV. In particular, they exhibited greater CSF speed, displacement, and net flow during regular breathing. Among respiratory factors, inhale length and diaphragm displacement showed the strongest correlations with CSF flow. Respiration modulated CSF dynamics through both mechanical enhancement of venous outflow and autonomic modulation of the heart, with mechanical effects predominating in the LV and both pathways contributing to the FM. Our findings identify respiration in the awake state as a modifiable, noninvasive mechanism that influences involuntary functions such as CSF circulation. This suggests that breathing may serve as a viable lifestyle intervention to support brain health by improving CSF-mediated homeostatic mechanisms, with potential future clinical applications in dementia prevention and care.

- **Feedback for revisions:**

Main: Last paragraph indicates methodology was examined in two groups (trained, T and non-trained group, NT). Yet, “Group-level comparison of baseline CSF dynamics”

begin with three groups (trained, healthy control, and AD)..Please clarify the study populations clearly in the Main last paragraph, and abstract.

=> Thank you for your thoughtful comment. We have redirected the focus of this study to comparisons between the T and NT breathing practice groups, as originally intended. All data from the previous publication, including AD participants and some healthy controls, have been removed from the analyses.

- **Feedback for revisions:**

Materials & Methods:

1. Please make it is easy for readers to understand the study population groups without reading several times. Study seems to include healthy participants which is further divided into two groups: trained (N=17) and non-trained (N=21), and also includes people with AD (N=10). But then following sections on M & M indicate including T=17 with 11 healthy controls and 10 AD. And Table S1 Demographics show T (17), HC (32), and AD (10). Total =59. And there is the HC=32 (NT + AD???)?

=> Thank you for your valuable comments. As suggested, we revised the overall structure of the manuscript to help readers better understand the study population without needing to reread multiple sections. Since the main focus of this study is the comparison between T and NT breathing groups, we have clearly defined these two groups throughout the manuscript and provided detailed demographic information in Table S1.

Please determine the right grouping, use associated acronym, either T, HC, and AD, or T, NT and AD, or HC (T, NT) and AD, which all depends on what healthy control is: whether healthy control vs AD or for T vs NT comparison.

Then please clarify each group's N# and in Table S1, apply modifications, and include mean (SD) age in years, age range in years, sex (provide also %) for each and every group after determining the right grouping. What's the p-value in Table S1 for? Please clarify.

=> Also, thank you for your thoughtful comments. The final study population includes trained participants (T, n = 20) and non-trained participants (NT, n = 25), with all AD participants and previously published data removed. In addition, we have updated Table S1 to include the group-specific mean age (with SD), age range, and sex distribution (%). Previously, the p-values in this table were difficult to interpret due to unclear grouping. With the revised grouping (T vs NT), the p-values now clearly indicate demographic differences between the two groups.

Table S1. Demographics

Demographic	T (n=20)	NT (n=25)	Difference (p-value)
Age			p=0.128
N (Nmiss)	20 (0)	25 (0)	
Mean (SD)	58 (17.3)	49 (20.2)	
Min - Max	29 - 81	21 - 82	
Sex			p=0.764
Female	8 (40%)	12 (48%)	
Male	12 (60%)	13 (52%)	
Systolic Blood Pressure			p=0.457
Mean (SD)	122.35 (9.30)	119.48 (14.91)	
Min - Max	104 - 143	97 - 159	
Diastolic Blood Pressure			p=0.140
Mean (SD)	74.15 (7.07)	78.36 (10.81)	
Min - Max	58 - 90	58 - 108	
Heart Rate, Regular Breathing			p=0.637
Mean (SD)	64.2 (8.04)	65.4 (8.91)	
Min - Max	52 - 79	48 - 89	
Heart Rate, Deep Breathing			p=0.699
Mean (SD)	66.0 (9.16)	67.1 (9.07)	
Min - Max	51 - 82	51 - 91	
HR RB vs DB (p-value)	p=0.006	p=0.094	
Respiratory Rate, Regular Breathing			p=0.013
Mean (SD)	8.4 (3.26)	14.1 (3.26)	
Min - Max	3 - 15	7 - 21	
Respiratory Rate, Deep Breathing			p=0.014
Mean (SD)	4.8 (2.33)	6.9 (3.06)	
Min - Max	2 - 10	3 - 18	
RR RB vs DB (p-value)	p<0.0001	p<0.0001	
Trained years, Mean (SD)			-
Mean (SD)	9.98 (8.98)	-	
Min - Max	1 - 27	-	

- **Feedback for revisions:**

Materials & Methods:

2. After determining right grouping, please provide inclusion/exclusion criteria for each group including AD group. Provide very brief information from previous study regarding recruitment of HC and AD, and what AD population neuropsychological exam entails without the need to refer to the previous study, ideally provide a Table S.

Since this study includes data from a previous study, please make it more clear in Participants section to determine which subject data is from the current study, and which from a previous study, and what exclusion and inclusion criteria for previous and new study were.

=> To avoid confusion, we have added a clear flow chart in the Supplementary Figure S1 and revised Table S1 to summarize the study groups and their inclusion/exclusion criteria.

- **Feedback for revisions:**

Materials & Methods:

3. "To ensure consistency, we selected a training method that focused explicitly on breathing practice and included only those who had engaged in a single, continuous program over an extended period."

Please explain what it means, and how was the inclusion criteria were determined? Participants were recruited based on what frequency/duration of their practice? e.g., 3 times/week 20 minutes, etc.? And with at least 1 year of practice? Practitioners with 1 year vs 30 years will have a wide range of variability in their results. How was this range clarified when analyzing data?

=> The T group included individuals with sustained formal training in the breathing practice of Seokmun Hoheup, each with more than one year of continuous practice. Participants with prior experience in other breathing practices were eligible for the T group provided that Seokmun Hoheup had been their primary and dominant practice for at least the past year. For recruitment, we collaborated with the Seokmun Hoheup Center, a non-profit organization, to identify practitioners with more than one year of consistent training who were interested in the study. While not a formal inclusion criterion, all T participants reported practicing at least one breathing practice session per week for approximately 90 minutes, and this has been clarified in the Methods.

The T group had a wide range of practice duration (mean 9.98 ± 8.98 years, range 1–27 years); however, CSF flow features were not associated with years of training,

suggesting that practice duration alone did not account for the observed group differences (Supplementary Figure S7).

We also added the breathing-related training history questionnaire to the study flow chart (Supplementary Figure S1). During screening, all participants (T and NT) were asked about prior experience with breathing practices (>6 months), the type of practice, duration (months/years), and frequency in the past 3 months (hours/week).

While we had a wide range of trained years average 9.98 (SD 8.98) ranging 1 to 27 years. 30 years was an error, it was actually 27 years. CSF flow features were not associated with years of training, suggesting that practice duration alone may not account for the observed group differences (Fig. S7).

Finally we have added the breathing-related training history questionnaire in the study flow chart (Figure S1). During screening, all participants, including both the T and NT groups, were asked the following questions:

1. *“Did you have any previous experience with breathing practices (more than 6 months), such as Seokmun Hoheup, diaphragmatic breathing, yoga, or mindfulness?”*
2. *“If yes, what kind of practice did you do?”*
3. *“If yes, how long have you been doing that practice? Please write in terms of months and/or years.”*
4. *“How often did you practice breathing in the past 3 months?”*

- **Feedback for revisions:**

- **Materials & Methods:**

4. “Participants in the NT group were screened using a questionnaire to confirm no prior sustained formal training in breathing practices (more than six months) such as diaphragmatic breathing, lower belly breathing, yoga, or mindfulness.”

How was this determined in terms of cut-off time close to the study? Was there a defined cut-off time, e.g., if someone has done 5 months breathing practice right prior to the beginning of the study, would they be included? In other words, if someone has concluded an 8-week MBSR prior to the study, would they be excluded? Please clarify inclusion/exclusion criteria as both the definition of long-term and duration of practice up until the start of the study even if it was 5 months (<6) is critical for complementary and integrative medicine modalities and their influence on study outcomes and/or clinically.

Considering many clinical trials include 8 to 12-week interventions to understand pre- and post-intervention outcome measures, a potential participant practicing for instance 3 months right until the start of the study would be excluded.

=> We thank the reviewer for this important comment. In this study, participants in the NT group were excluded if they had engaged in any structured breathing practice for more than six months. Thus, we recruited individuals who had not practiced any structured breathing method during the six months prior to study enrollment and who also had no prior history of structured training exceeding six months.

In theory, cases such as five months of recent practice or completion of an 8–12 week MBSR program could fall into a gray area. However, we did not encounter such cases during recruitment. If they had arisen, we would not have included those participants in the NT group in order to maintain a strict definition of “non-trained.” This consideration has been clarified and noted in the Discussion.

(added in the Discussion) This study has several limitations. First, “non-trained” participants were defined broadly, without considering shorter-term structured breathing programs, which have been shown to modulate physiological outcomes (Ref: Grossman, P., Niemann, L., Schmidt, S. & Walach, H. Mindfulness-based stress reduction and health benefits. A meta-analysis. J Psychosom Res 57, 35-43 (2004).).

- **Feedback for revisions:**

Materials & Methods:

5. “The practice combines physical movement with focused attention on a specific point in the lower abdomen to encourage natural, sustained breathing habits.”

Please define physical movement whether this is movement of abdomen alone or actual synchronized physical motions along with breathing like in Tai-Chi, Qi-Gong, etc.

=> We thank the reviewer for this question. Seokmun Hoheup is a structured and systematic program that incorporates multiple components of gentle exercise coordinated with breathing. A detailed description of the program and its components has been included in the Methods section.

Method Section

“Each training session lasts approximately 90 minutes and consists of four components: a ‘warm-up exercise’ (Chejo, 10 min), ‘Haenggong’ (24 min), ‘breathing practice’ (48 min), and ‘recovery stretching’ (Hoegeonsul, 12 min). The ‘warm-up exercise’ involves gentle body movements to stabilize physiological functions, relax muscles, and prepare

the body and mind for the practice. 'Haenggong' comprises 12 whole-body postures, each maintained for two minutes, to facilitate transition into the core breathing practice. The 'breathing practice' is performed in different positions (lying down, sitting, or standing) depending on the training stage. Sessions conclude with 'recovery stretching', which consists of gentle movements synchronized with breathing to promote relaxation and improve circulation.

Through repeated experiential training, Seokmun Hoheup practice emphasizes body awareness, postural stability, and activation of the diaphragm, fostering fine, long, and deep breathing with an inhale-to-exhale ratio typically around 6:4. In this study participants in the T group performed the 'breathing practice' component of these sessions."

- **Feedback for revisions:**

Materials & Methods:

6. For a smooth read: Please remove CSF measurement related sections within Participants section and move to a new CSF measurements subheading or move to MRI Acquisition sections. It is confusing to read the paper with its current form. Change Breathing Protocol to MRI Breathing Protocol to differentiate Breathing Training vs Breathing practices performed during MRI acquisitions. Overall, please revise the experimental methodology sections to ensure a smooth and cohesive flow.

=> We thank the reviewer for this helpful suggestion. We have revised and rearranged the methodology sections accordingly to address this comment.

- **Feedback for revisions:**

Materials & Methods:

7. Include a study flow chart (e.g., similar to consort diagram, including number and reasons of excluded participants from each group).

=> We thank the reviewer for this helpful suggestion. We have added a study flow chart (Supplementary Figure S1) to illustrate participant inclusion and exclusion, group designation, and final group totals.

❖ Reviewer #2 (Remarks to the Author):

● To the authors:

This study provides a meaningful and timely contribution to the understanding of cerebrospinal fluid (CSF) dynamics, particularly by highlighting the potential impact of breathing methods on CSF inflow via autonomic modulation of the cardiovascular system. The application of multiple imaging techniques to explore CSF dynamics at the foramen magnum and lateral ventricles is commendable, and the focus on diaphragmatic movement introduces an important physiological perspective that broadens the scope of CSF research. However, the classification of experimental groups according to breathing practices—arguably the core variable of this study—would benefit from further clarification and scientific rigor. The current design limits the interpretability of key findings. The reviewer suggests the authors consider the following points to strengthen the scientific validity and overall impact of their work:

● Major Comments:

1. Group classification and Variable Control: The current categorization of the “Training” group based on a single breathing technique (Seokmun breathing) lacks a precise scientific definition and variable control. For a more rigorous approach, we recommend identifying and measuring key physiological parameters that differ between trained and control subjects (e.g., respiratory rate, tidal volume, I:E ratio). Analyzing how each of these factors independently and interactively affects CSF dynamics would improve mechanistic understanding. For example, a quantitative correlation between diaphragmatic movement and other physiological variables could clarify causal pathways.

=> We thank the reviewer for this important comment. The Training group was defined as participants with >1 year of consistent Seokmun Hoheup practice, but to address the reviewer’s concern we analyzed key physiological variables directly rather than relying solely on group categorization.

In the revised manuscript, we emphasize Figure 2 as a central result, where pathways from respiratory features to CSF flow features were modeled using structural equation modeling (SEM) and multivariate regression. At the FM, CSF net flow was influenced by both mechanical (SSS displacement) and autonomic (HR displacement) pathways, while at the LV CSF dynamics were predominantly driven by the SSS pathway. Among respiratory variables, inhale length and diaphragm displacement emerged as key contributors.

Following the reviewer’s recommendation, we expanded the respiratory features analyzed to include breathing length (respiratory rate), lung volume change (tidal

volume), the I:E ratio, chest displacement, and diaphragm displacement. Breathing length showed the strongest correlation with CSF dynamics (Supplementary Fig. S9) and was decomposed into inhale and exhale length, thereby incorporating the I:E ratio. Independent effects of each feature are presented in Supplementary Fig. S9. The SEM model demonstrates interactive effects, with inhale length associated with SSS venous displacement (blood outflow) and diaphragm displacement associated with HR modulation (blood inflow), consistent with the Monro-Kellie principle of CSF, arterial, and venous balance.

- **Major Comments:**

2. Autonomic and Cardiovascular Evidence: While the hypothesis regarding autonomic modulation of the cardiovascular system is conceptually appealing, direct evidence—such as group-wise differences in HR, BP, and autonomic markers (e.g., catecholamines, cortisol)—was not presented. Future studies could strengthen this argument by evaluating whether similar CSF changes are observed in other autonomic conditions (e.g., during exercise or sleep) to validate the generalizability of the proposed mechanism.

=> We thank the reviewer for this insightful comment. We agree that we did not include direct autonomic or cardiovascular measurements (e.g., blood pressure, catecholamines, cortisol) in the present study. This is a limitation of the current work. Future studies are planned to integrate such autonomic and cardiovascular measures alongside MRI-based CSF assessments to more fully evaluate these mechanistic pathways. We also agree that testing whether similar CSF changes occur in other autonomic contexts (e.g., exercise, sleep, or pharmacologic modulation) will be important to validate and generalize the proposed mechanism. A paragraph has been added to the Discussion to acknowledge this limitation and outline these future directions.

(Added paragraph in the Discussion section):

“This study has several limitations. First, “non-trained” participants were defined broadly, without considering shorter-term structured breathing programs, which have been shown to modulate physiological outcomes. Second, we did not include direct autonomic or cardiovascular measurements such as real-time blood pressure, heart rate variability, or biochemical markers (catecholamines, cortisol), limiting interpretation of autonomic contributions. Future studies incorporating multimodal monitoring and testing across other autonomic conditions (e.g., exercise, sleep, pharmacologic interventions) will be needed to validate and generalize these mechanisms.”

- **Major Comments:**

3. Diaphragmatic Movement vs. CSF Dynamics: Although diaphragmatic movement is discussed as a critical factor, some data suggest notable CSF differences despite minimal differences in diaphragm displacement between NT and T groups. Addressing this apparent discrepancy would provide greater insight into the underlying physiological mechanisms.

=> We thank the reviewer for this thoughtful observation. We agree that diaphragmatic displacement alone does not fully explain the group-wise differences in CSF dynamics. In our revised analysis, inhale length emerged as an effect independent from diaphragm displacement (Figure 2 and Supplementary Figure S9). Our interpretation is that while the spatial change of diaphragmatic movement is important, the temporal aspects of breathing, particularly inhale length, may play a critical role in modulating CSF flow. We have added further discussion and clarification in the revised manuscript regarding the contributions of these respiratory features to CSF dynamics.

(Added paragraph in the Discussion section):

“These group-level differences were accompanied by longer inhale length and greater diaphragm displacement in trained participants, particularly during regular breathing, reflecting temporal and spatial features of respiration that were closely coupled with CSF dynamics (Fig. 3). Extending beyond group comparisons, analysis across all participants showed that higher CSF net flow was linked to longer inhale length, greater diaphragm displacement, and higher SSS displacement, underscoring the importance of both temporal and mechanical aspects of respiration in driving CSF circulation (Fig. 4).”

- **Major Comments:**

4. Baseline Physiological Metrics: Given that trained individuals exhibited higher mean absolute CSF velocities at rest, reporting baseline cardiovascular and respiratory variables (e.g., HR, RR) is essential to rule out pre-existing physiological differences as confounding factors.

=>We thank the reviewer for this point. Baseline heart rate during both RB and DB showed no significant differences between groups. Respiratory rate, however, was significantly lower in the trained group at both RB and DB, which we think is consistent with long-term breathing practice. These results are now presented in Table S1. To address the reviewer’s concern regarding potential confounding, we think both HR and respiratory rate should be considered key variables, and they were incorporated into our SEM and multivariate analyses (Fig. 2a). In this framework, they are treated not as uncontrolled confounders but as mechanistic features of the training effect, allowing us to directly test their contribution to CSF dynamics.

- **Major Comments:**

5. Intra-individual Breathing Comparison: A direct comparison of resting vs. deep breathing within the same individuals—particularly in both trained and untrained groups—would greatly enhance the interpretability of the training effect and help isolate the influence of breathing depth.

=> We thank the reviewer for this valuable suggestion. We have added Figure S5, which presents intra-individual comparisons of regular versus deep breathing in both trained and untrained groups.

(Added in revised Result section)

“Within-subject comparisons of RB and DB (Fig. S5) showed that CSF displacement increased with DB in both the FM and LV across T and NT participants (T-FM: $p = 0.0003$; NT-FM: $p < 0.0001$; T-LV: $p = 0.0470$; NT-LV: $p < 0.0001$). In contrast, CSF net flow increased with DB only in the FM (T-FM: $p = 0.0199$; NT-FM: $p = 0.0024$)”

- **Major Comments:**

6. Correlation Analysis Between Groups: To support the claim of a positive correlation between HR displacement and lateral ventricle (LV) CSF displacement, correlation analyses should be performed separately for the trained and untrained groups, ensuring group-specific effects are not conflated.

=> We thank the reviewer for this helpful comment. In the revised manuscript, we have shown separate correlation analyses for the trained and untrained groups. Specifically, two regression lines are displayed for each group in all of our Pearson correlation plots.

- **Minor Comments:**

1. Use "mean speed" instead of "mean velocity" when referring to the absolute value of velocity.

=> We thank the reviewer for this comment. We have revised the manuscript to use “mean speed” instead of “mean velocity” when referring to the absolute value of velocity.

- **Minor Comments:**

2. Lines 166–170 are speculative and not directly supported by the presented data; there is no clear link between CSF inflow changes and glymphatic function.

=> We thank the reviewer for this helpful comment. We agree that our original statement was speculative and not directly supported by the presented data. We have revised the text to clarify that while previous studies suggest autonomic activity may influence glymphatic function, our results only demonstrate respiration-related modulation of CSF dynamics during wakefulness. We now frame this point as contextual discussion rather than a direct conclusion.

(Added paragraph in the Discussion section):

“Previous studies suggest that autonomic activity may influence glymphatic function, with sympathetic tone suppressing and parasympathetic tone enhancing glymphatic flow. While we did not directly measure glymphatic clearance, our results demonstrate that respiration modulates CSF dynamics during wakefulness, which may provide a physiological context for future studies of glymphatic function. Consistent with this, a recent study demonstrated that enhancing RSA through paced breathing reduced plasma amyloid- β and tau levels in both younger and older adults, suggesting that autonomic regulation of respiration may contribute to lowering neurodegenerative risk.”

- **Minor Comments:**

3. The statistical analysis section requires more detail, including complete p-values in all figure panels. For example, Figures 3f and 3g lack p-values.

=> We thank the reviewer for this comment. All figure panels have now been updated to include complete p-values

- **Minor Comments:**

4. Figure 2: To improve clarity, present correlation statistics separately for the T and NT groups.

=> We thank the reviewer for this helpful suggestion. In the revised manuscript, we now present all Pearson's correlation statistics separately for the T and NT groups to improve clarity.

- **Minor Comments:**

5. Please verify the P-value in Table S1—is 0.000 statistically appropriate?

=> We have updated Table S1.

- **Minor Comments:**

6. Re-check the P-value in Supplementary Figure 2f. The legend states a significant difference, yet the P-value appears to be 0.512.

=> Done. Thanks.

- **Minor Comments:**

7. Include individual data points (dots) on all bar graphs to enhance data transparency.

=> Done. Thanks.

- **Minor Comments:**

8. In figures such as S6b/d and S7b/d, redraw the y-axis starting from 0, or use break symbols if a truncated axis is needed to show variation clearly.

=> Done. Thanks.

- **Minor Comments:**

9. Indicate the number of individuals included in Fig S6b in the legend.

=> The model was separately trained on 90 recordings acquired from 45 individuals during both deep and regular breathing conditions (n = 90).

- **Minor Comments:**

10. Figure 3a is not referenced in the manuscript.

=> Thanks.

- **Minor Comments:**

11. Justify the extraction of voxels related to cardiac signals during lateral ventricle analysis.

=> We thank the reviewer for this comment. Because CSF dynamics are known to be influenced by cardiac components, we applied the same ROI segmentation algorithm used in other regions to the LV in order to test whether cardiac-related signals were also detectable there. This approach ensured consistency of analysis across regions and does not bias our respiratory-focused investigation of CSF dynamics.

- **Minor Comments:**

12. Line 200: Please confirm whether "LV displacement" is the correct term.

=> We have update it to LV CSF displacement.

- **Minor Comments:**

13. Line 203–204 and Figure 3e: Clarify which group the trained individuals were compared to and how the stated P-value was derived.

=> The figure has been reorganized to improve clarity. In the revised version, it is now clearly indicated that trained individuals were compared to the non-trained group, and the stated p-values were derived using unpaired two-tailed Student's t-tests.

- **Minor Comments:**

14. Add P-values to Figures S9a and S10a to improve readability.

=> We thank the reviewer for this suggestion. These figures are now Figure S6. In the revised manuscript, we marked a star on the plots where statistical significance shown with mean \pm 95% confidence interval.

- **Minor Comments:**

15. Clarify the CSF net flow patterns at the foramen magnum and lateral ventricle by adding sub-headers or detailed labels in relevant graphs.

=> Done. Thanks.

- **Minor Comments:**

16. The order of supplementary figures does not follow the manuscript's flow (e.g., Supplementary Figure 11); please revise accordingly.

=> In the revision, we have made the supplementary figure follow the manuscript flow. Thanks.

- **Minor Comments:**

17. The hypothesized role of the superior sagittal sinus (SSS) as a driver of CSF flow needs mechanistic support. Consider including velocity-encoded MRI data, particularly after Q-collar application, to support this claim.

=> We thank the reviewer for this suggestion. Although the Q-collar data were interesting, the sample size was limited ($n = 5$), and the velocity-encoded MRI data did not show statistical differences at this scale. We had to apply bootstrapping to build the model, and given these limitations, we have removed the Q-collar results from the revision. We plan to expand this dataset and present it as part of a future, separate paper.

- **Minor Comments:**

18. Provide the P-value for Figure 5b.

=> We thank the reviewer for this comment. Figure 5 has been removed in the revised manuscript, as inhale length feature explained most of the content originally presented in this figure.

- **Minor Comments:**

19. Figure 6 appears to present new data but is only discussed in the Discussion section. Consider relocating this to the Results section for coherence.

=> We thank the reviewer for this comment. Figure 6 has been removed in the revised manuscript, as it was based on a smaller cohort sample and did not add substantial value to the main findings.

❖ **Reviewer #3 (Remarks to the Author):**

In the manuscript by Lim et al, the authors propose that controlled breathing exercises can enhance CSF fluid flow within the lateral ventricles. The authors use MRI techniques on populations of trained and untrained subjects during breathing, in an attempt to measure differences in CSF flow between these groups. The authors proposed that breathing exercises may be a lifestyle intervention to improve CSF circulation and potentially waste clearance. While this is an attractive and novel hypothesis, the data in the current manuscript does not support this concept.

● **Major comments**

1. The title does not reflect the content. The most pronounced effects of respiration on CSF dynamics occur at the foramen magnum rather than the lateral ventricle. There are no meaningful significant effects on CSF displacement or flow between non-trained and trained individuals in Figure 3b or e. I also fail to see a significant effect of deep breathing on these parameters compared to regular breathing in either group.

=> We thank the reviewer for this thoughtful comment. We initially emphasized the lateral ventricle (LV) in the title because we found it intriguing and important that respiratory effects could be observed in the deep brain CSF area. However, we agree with your perspective. We acknowledge that the foramen magnum (FM) provides more significant results, showing changes in all CSF features related to respiratory parameters, while the lateral ventricle (LV) provides secondary results with more limited respiratory effects. One of our main conclusions is that the FM, located at the entry of the skull, and the LV, representing deep brain CSF, demonstrate differential respiratory influences (Figure 2). To better reflect this, we have revised the title to emphasize the respiratory effect on the underexplored CSF feature of net flow: "Human Cerebrospinal Fluid Net Flow Enhanced by Respiration During the Awake State."

Regarding the lack of robust training-related group differences in LV CSF features, as shown in Figure 1, we note that during regular breathing, differences between trained and non-trained participants are observed in all three LV CSF measures. These differences diminish during deep breathing. One potential interpretation is that baseline regular breathing already differs between the groups, as shown in Figure 3, where trained individuals exhibit longer inhale length and greater diaphragm displacement. This suggests that breathing training may shift baseline breathing patterns, which in turn influence CSF dynamics.

In addition, Figure 4 shows that when participants are divided into tertiles based on FM net flow, inhale length and diaphragm displacement differ significantly, and SSS venous

displacement also shows a difference. Interestingly, even when grouped by FM net flow, LV CSF displacement and net flow also differ between the highest and lowest tertiles, suggesting that FM-driven differences may propagate to deep brain CSF measures.

We hope these findings highlight meaningful regional differences and provide novel insights into how respiration influences CSF dynamics at both the FM and LV.

- **Major comments**

2. The authors appear to have reused previous data in the study. The paper integrates 11 healthy controls from a prior published study and rebrands them as part of a healthy control (HC) group with the new NT cohort stating, “we combined the 11 previously studied healthy participants with the 21 NT participants... and labelled this combined group as the healthy control (HC) group.” This pooled group lacks internal consistency, as the reused participants did not undergo the same protocols (e.g., deep breathing or diaphragm MRI), yet are directly compared. This practice inflates sample size and may confound interpretations of CSF flow baselines. The inclusion of Alzheimer’s disease patients from this earlier study is also problematic. There is no reasoning given why these patients are included in the current study. The data from these patients is included in Fig 1 and Fig 4, but not in other figures.

=> We thank the reviewer for this important comment. We acknowledge that including participants from a prior study could create concerns regarding protocol consistency and interpretation. To address this, we have removed the previously published healthy control data as well as the Alzheimer’s disease data from the current analyses. The revised manuscript now focuses exclusively on the primary comparison between the trained (T) and non-trained (NT) groups, using only participants who underwent the full, consistent protocol in this study.

- **Major comments**

3. The manuscript does not clearly separate the methods, results, and discussion sections, often mixing explanations of the analysis with descriptive text. For example, references to “Fig. 4” and “the final causal diagram” appear in places where it is unclear whether the authors are describing their procedures or interpreting the findings. Some figures are introduced in the discussion, such as Figure 6. Also, the important flow phantom validation, which should be described in the methods, is only mentioned late in the discussion. Supplementary Figure S16 is never referenced or explained in the main text. Together, this disorganisation reduces the clarity the paper, making it difficult to follow the study’s narrative.

=> We thank the reviewer for these comments. In the revised manuscript, we have reorganized the Methods, Results, and Discussion sections to improve clarity and narrative flow. The Methods now clearly describe all experimental procedures and analyses, including the flow phantom validation, while the Results are focused on reporting the findings without interpretive overlap. We have also ensured that all figures and supplementary materials, including Supplementary Figure S16 (now revised as Figure S12), are properly referenced and explained in the main text.

Reviewer #1

Remarks to the Author:

Summary

As indicated in initial review, the central question of this manuscript is timely and important, with strong implications for advancing integrative health research within the context of CNS health. The findings are potentially impactful and of broad interest. The authors have made significant progress in addressing previous review comments. However, further revisions are needed to strengthen the manuscript. Please see the critiques and comments below (rather long), that are intended to improve the clarity, rigor, and accessibility of the work, all to ensure study contributions are communicated in the strongest possible way.

=> We thank the reviewer for their thoughtful assessment and for recognizing the importance and potential impact of this work. We appreciate the acknowledgement of the progress made in revision and will address the remaining points in detail below to further strengthen and clarify the manuscript.

Major Comments

Methods/Study Flow: The overall experimental approach and methodology are appropriate (except see below for a major concern). The description of methods, however, could be improved for readability via presenting them in a more chronological order that follows the actual study workflow. This would help readers better follow the logic of study design and data collection and remove remaining confusion from initial manuscript.

=> We thank the reviewer for this thoughtful suggestion. In the revised manuscript, the Methods section has been reorganized to follow the chronological order of the study workflow. In addition, the description of the phantom validation study has been moved to the beginning of the Methods section to present the technical validation prior to the *in vivo* experiments.

Choice of CSF Metrics: The authors report “mean CSF speed” as a primary outcome, rather than peak, minimum, or peak-to-peak CSF velocity values. This choice raises concerns about interpretability. Because CSF dynamics are characterized primarily by pulsatility and oscillations, mean values may obscure physiologically meaningful fluctuations.

=> We thank the reviewer for this important comment and the opportunity to clarify our rationale. We recognize the physiological relevance of peak, minimum, and peak-to-peak velocity values, as CSF dynamics are inherently oscillatory. These peak-based metrics showed strong correlations with one another; therefore, in the revised manuscript, we incorporated the mean peak velocity features into Figures 1 and 4 to represent overall velocity characteristics. Importantly, mean peak and mean valley velocities were already evaluated in Supplementary Figure S10, where both demonstrated very high correlations with CSF mean speed. As previously reported by our group¹, CSF mean speed provides a stable representation of overall CSF flow

magnitude and serves as a robust, physiologically informative measure for distinguishing normal and pathological conditions. To further support this, we added a new supplementary analysis comparing maximum, minimum, peak-to-peak velocity, and mean speed. The terminology has also been updated throughout the manuscript from “Mean CSF speed” to “CSF mean speed” for consistency.

- The authors should clarify why mean CSF speed was prioritized. In particular, why was speed (magnitude only) chosen instead of velocity, which preserves directionality and is more physiologically informative?

=> One of the initial goals of this study was to investigate the directional influence of respiration on CSF flow. Previous studies have reported a consistent pattern, with CSF moving rostrally during inhalation and caudally during exhalation^{2,3}, while directionality has also been shown to vary across different anatomical regions of the spine and cranial base⁴. In our preliminary analysis, we identified inhalation and exhalation periods within each scan and calculated CSF velocity (cm/s) and displacement (ml) for each phase of respiration (Shown here). Interestingly, we observed that directional patterns sometimes varied with breathing length and across participants. This variability could reflect interindividual differences in respiratory phase timing or the inherent constraints of 2D PC-MRI in capturing multidirectional flow. Future studies employing multidirectional velocity encoding, such as 4D flow MRI, will be needed to further investigate directional characteristics.

Figure 1. Directional CSF volume displacement calculation. The CSF Volume (ml) traces show directional volume changes during inhalation (pink) and exhalation (purple) periods. Contrary to our expectation, a consistent directional pattern was not observed across participants.

In this study, we therefore focused our analysis on the amplitude (magnitude) of CSF motion rather than its directionality. Our previous real-time 2D phase-contrast MRI work demonstrated that the magnitude of velocity amplitude was most informative for distinguishing physiological and pathological conditions¹. Building on this, the current

study uses mean speed as one of our main measurements to quantify the amplitude of CSF movement, capturing cardiorespiratory influences independent of flow direction. Although directionality remains an important parameter, we speculate that respiration-driven directional changes are more subtle than cardiac-driven ones, whereas amplitude-based features such as mean speed and displacement may be more reliably reflect cardiorespiratory modulation of CSF motion. Thus, our focus on mean CSF speed highlights amplitude-driven dynamics that are sensitive to cardiorespiratory influences, providing a consistent and physiologically meaningful measure across participants.

We included a discussion point acknowledging that directionality is an important aspect of CSF dynamics but was not assessed in the current paper.

“Directionality is an important aspect of CSF dynamics. Previous studies have reported a general pattern, with CSF moving rostrally during inhalation and caudally during exhalation ^{2,3}, and have shown that this pattern can vary across anatomical regions of the spine and cranial base ⁴. In the present study, we focused on the magnitude of CSF motion rather than its directionality. Amplitude-based measures were used to characterize overall CSF motion and its cardiorespiratory influences using 2D real-time PC-MRI. Future studies using 4D flow MRI could provide a more comprehensive assessment of respiratory effects on CSF directionality.”

- Since the study investigates the effects of regular versus deep breathing on CSF dynamics, peak (maximum, minimum, and/or peak-to-peak) velocity measures are more valuable for quantifying the true impact of breathing on pulsatile CSF. Including such measures would also allow clearer interpretation of differences across breathing patterns (regular vs. deep) and participant groups (T vs. NT), and would align better with the physiological rationale while capturing true dynamics (e.g., modulation of CSF by diaphragm motion).

=> We thank the reviewer and fully agree with the suggestion. As described above, the peak-based velocity values have been integrated into the main figures.

The choice of additional CSF metrics -esp. in correlation analysis- appears somewhat selective, and the rationale for their inclusion vs. other commonly used measures is not fully clear.

=> We thank the reviewer for requesting clarification. We have added a table summarizing all CSF flow features included in this study. The flow analysis progressed stepwise from velocity (cm/s) to flow rate (ml/s) and finally to volume (ml), with specific features derived at each stage. Additionally, the original Supplementary Figure S8 has been moved to Supplementary Figure S5 to illustrate the relationships among all features and to clarify the overall analytical framework. To further aid understanding, we provide below a concise summary of our CSF feature extraction and quantification workflow.

An important point is that our data were not derived from any bandpass-filtered signal; rather, the analysis used the fully intact CSF velocity time series throughout the entire feature extraction process. The algorithm was adapted from Liu et al. (2025), which has been validated in real-time PC-MRI CSF flow studies^{3,5}.

Based on the well-established concept that CSF velocity is primarily driven by cardiac pulsations, whereas CSF displacement is predominantly influenced by respiratory modulation^{3,6,7}, our rationale was to investigate whether these features change with breathing condition (regular vs. deep) and differ between trained and non-trained groups, and further whether these effects persist in the derived net flow measures.

- ROI Selection: ROIs were identified by selecting voxels exhibiting strong cardiac-driven oscillations (0.8–2.0 Hz) using voxel-wise FFT analysis. Because cardiac-driven CSF oscillations are robust and reproducible, this approach ensured consistent ROI definition while minimizing potential bias when assessing respiration-related CSF effects.

- Data Extraction: The ROI mask was then applied to the raw PC-MRI data to extract the velocity time course without filtering, and linear detrending was performed to remove DC offsets and slow drifts.

- Velocity Features: Peak detection identified systolic (peak) and diastolic (valley) velocities to compute mean peak velocity (cm/s), mean valley velocity (cm/s), and mean PV (cm/s) (average of peak and valley). The mean CSF speed (cm/s) (mean of absolute velocity) quantified total CSF movement independent of flow direction.

- Flow Rate: To normalize for ROI size differences across scans, velocity was converted to flow rate (ml/s) by multiplying mean velocity by voxel area, slice thickness, and voxel count, generating a CSF flow rate time course.

- Volume Conversion and Respiratory-Coupled Features: Integration of this flow rate yielded the CSF volume displacement time course (ml). Using simultaneously recorded respiratory belt data, each respiratory cycle was identified and used to calculate CSF maximum and minimum flow rate, CSF displacement (difference between maximum and minimum displacement per respiratory cycle), and CSF net flow (absolute sum of directional displacement).

This comprehensive framework ensures that all features were derived systematically and physiologically motivated, aligning with established findings on cardiorespiratory-driven CSF dynamics.

Table S2. Summary of Features

Source Data (unit)	Cycle reference	Features computed (unit)	Used in
CSF velocity time course (cm/s)	None	- CSF mean peak velocity (cm/s)	Fig. 1, Fig. 4, Fig. S4, Fig. S6, Fig. S7, Fig. S8
		- CSF mean valley velocity (cm/s)	Fig. S7, Fig. S8
		- CSF pk-to-pk	Fig. S7
		- CSF mean PV (cm/s)	Fig. 2a, Fig. S10
		- CSF mean speed (cm/s)	Fig. 1, Fig. 2b, Fig. 4, Fig. S4, Fig. S5, Fig. S9, Fig. S10, Fig. S11, Fig. S12
CSF flow rate time course (ml/s)	Respiratory	- CSF maximum flow rate (ml/s)	Fig. 2a, Fig. S10
		- CSF minimum flow rate (ml/s)	Fig. S10
CSF volume displacement time course (ml)	Respiratory	- CSF displacement (ml)	Fig. 1, Fig. 2a, Fig. 2b, Fig. 4, Fig. S4, Fig. S5, Fig. S9, Fig. S10, Fig. S11, Fig. S12
		- CSF net flow (μ l)	Fig. 1, Fig. 2a, Fig. 2b, Fig. 4, Fig. S4, Fig. S5, Fig. S9, Fig. S10, Fig. S11, Fig. S12
SSS venous flow rate time course (ml/s)	Respiratory	- SSS venous max flow rate (Resp) (ml/s)	Fig. S16
SSS venous volume displacement time course (ml)	Respiratory	- SSS venous displacement (ml)	Fig. 2a, Fig. 3, Fig. 4, Fig. S10, Fig. S12, Fig. S13, Fig. S16
		- SSS venous displacement (Resp) (ml)	Fig. S16
		- SSS venous displacement (Cardio) (ml)	Fig. S16
PPG	Cardiac	- HR (BPM)	Fig. S13, Fig. S16
		- HR displacement (Δ BPM)	Fig. 2a, Fig. 3, Fig. 4, Fig. S10, Fig. S12, Fig. S13, Fig. S16

Respiratory belt	Respiratory	- Respiratory rate (Breaths/min)	Fig. S6, Fig. S11, Fig. S13
		- Breathing length (sec)	Fig. S11
		- Inhale length (sec)	Fig. 2a, Fig. 2b, Fig. 3, Fig. 4, Fig. S10, Fig. S11, Fig. S13, Fig. S16
		- Exhale length (sec)	Fig. S10, Fig. S11, Fig. S13, Fig. S16
		- Inhale:Exhale Ratio	Fig. S11, Fig. S13, Fig. S16
Diaphragm motion MRI	Respiratory	- Lung area displacement	Fig. 2a, Fig. S10, Fig. S11, Fig. S13, Fig. S16
		- Chest displacement	Fig. 2a, Fig. S10, Fig. S11, Fig. S13, Fig. S16
		- Diaphragm displacement	Fig. 2a, Fig. 2b, Fig. 3, Fig. 4, Fig. S10, Fig. S11, Fig. S13, Fig. S16

Lastly, as a general note, there should be a greater clarity and consistency in the terminology used for the CSF measure. What PC-MRI measures is the velocity of the CSF pulsatile motion/movement, and secondary metrics like CSF net flow and displacement are derived from it. Yet often studies may use the term CSF flow or circulation. While authors may have their own framing, it would be important to adopt a consistent use of a term: CSF dynamics or pulsatile CSF motion or oscillatory movement, which will strengthen the manuscript, and will also help avoid confusion for readers.

=> We thank the reviewer for this important comment. We agree that clarity and consistency in terminology are critical. We have standardized the manuscript to use two complementary terms. Specifically, we use “CSF movement” to describe the velocity-derived local measures (mean speed, displacement, and net flow) directly obtained from the PC-MRI data, and “CSF dynamics” to describe the broader physiological framework that encompasses these measures.

We have also added the following sentence in the Methods section for clarity:

“In this manuscript, we use the term CSF movement to refer to the velocity-derived local measures (mean speed, displacement, and net flow), and we use CSF dynamics to encompass the broader concept of cerebrospinal fluid behavior.”

Other comments:

Abstract:

1. Please use “mean (\pm SD)” in age related sections, e.g.: mean (\pm SD) age 58.1 \pm 17.3 years.

=> Done. Thanks.

2. “All underwent real-time velocity-encoding magnetic resonance imaging (MRI) to assess CSF flow at the foramen magnum (FM) and lateral ventricle (LV).”

Please add “during regular and deep breathing” to provide methodological information upfront.

=> Done. Thanks.

3. “Our findings identify respiration in the awake state as a modifiable, noninvasive mechanism that influences involuntary functions such as CSF circulation.”

Authors in this study compute “CSF speed, net flow and displacement”. It is important to note that displacement reflects oscillatory and bidirectional motion rather than unidirectional transport. Similarly, net flow reflects directional bias in CSF movement, may be small, and not sustained enough to represent circulation. CSF circulation on the other hand involves systematic / global turnover through ventricular system and SAS, and requires turnover pathways rather than local oscillatory motion dynamics. Thus, the use of “CSF circulation” in above statement is not supported by the findings.

=> We agree with the reviewer’s comment. The term “CSF circulation” has been replaced with CSF movement or CSF dynamics throughout the manuscript for accuracy and consistency.

4. “This suggests that breathing may serve as a viable lifestyle intervention to support brain health by improving CSF-mediated homeostatic mechanisms, with potential future clinical applications in dementia prevention and care.”

While breathing is known to regulate CSF dynamics, mechanistic pathways between brain health and respiratory driven CSF-mediated homeostatic processes remain unclear. Further, current research does not establish how such mechanisms could be linked to dementia prevention, and authors are not examining CSF-ISF driven glymphatic clearance but rather investigating the impact of breathing in long-term breathing practitioners vs. novices. Thus, while the suggestion above may be plausible, it is not sufficiently supported by the evidence presented. It can be, and in my opinion, certainly shall be included in discussion.

=> We agree with the reviewer’s comment. The abstract has been revised to remove speculative statements about dementia prevention. The conclusion now focuses on the observed effects of respiration on CSF dynamics and homeostatic regulation.

5. Please also include in the results: how regular vs. deep breathing findings differ among practitioner vs. novice participants.

=> Thank you. The abstract has been updated to include a summary comparing regular and deep breathing findings between trained and non-trained participants.

The revised section now reads:

“DB enhanced CSF dynamics in both groups, increasing displacement and net flow, particularly at the FM. Seokmun Hoheup-trained participants showed greater CSF movement than controls at both the FM and LV. Even during RB, trained participants showed higher CSF mean speed, displacement, and net flow.”

Introduction:

1. References included in introduction can be reviewed for a revision as some of the citations may not be representing the statements, while some statements do not include the full list of citations included in references.

=> Thank you. We have carefully reviewed the references in the Introduction and updated them to ensure that each citation accurately represents the corresponding statements and that all relevant references are appropriately included.

2. Similarly, as in abstract, please consider modifying

“All participants underwent CSF flow measurement in the brainstem and the lateral ventricles using velocity-encoding MRI during both resting wakefulness and voluntary slow, deep breathing.” as

“All participants underwent CSF flow measurement in the brainstem and the lateral ventricles using velocity-encoding MRI during regular breathing and voluntary slow, deep breathing, both acquired during wakefulness” or something on that order to mention regular and deep breathing terminology upfront.

=> Done. Thanks.

The revised section now reads:

“All participants underwent CSF flow measurement at the foramen magnum (FM), located at the craniospinal junction, and in the lateral ventricle (LV) using velocity-encoding MRI during both regular breathing (RB) and voluntary slow, deep breathing (DB) acquired during wakefulness.”

3. In last paragraph, please consider including a line for phantom measurements for validation of the MRI technique prior to indication of examining two groups.

=> Thank you. The revised section now reads:

“In this study, we first validated the velocity-encoding MRI technique using a flow phantom to ensure measurement accuracy and reproducibility. We then examined two groups:...”

Results:

1. Figure S1. Inclusion of questionnaire in study flow chart seems wrong. Perhaps it was meant to be included prior to enrollment/consent process since those within the NT group would be determined prior to enrollment based on questionnaire.

=> Thank you. We have revised the flow chart to better reflect the study order.

2. p values throughout the paper shall be included as $p=X$ or $p<X$ rather than $p = X$, or $p < X$.

=> Thank you. Done

3. Use of acronyms for the rest of the manuscript after it was defined for consistency, such as HR for heart rate, RB for regular breathing, DB for deep breathing, etc... once it was defined.

=> Thank you. We have double-checked all acronyms and ensured consistency throughout the manuscript.

4. Please clarify the use of speed instead of velocity in human experiments, while the term velocity is used for phantom experiments. And as in also noted above & in Methods section critiques, please clarify use of mean CSF speed instead of max and min, or peak-to-peak CSF velocity for a study quantifying the impact of breathing during distinct breathing patterns, to capture dynamic ranges.

=> Thank you. We agree that peak-based velocity features are important features of CSF dynamics. We have integrated them in our main data. The flow phantom data were analyzed using the same approach, and the terminology has now been updated to “CSF mean speed” for consistency across both the human and phantom studies.

5. While the T group shows higher mean CSF speed values compared to the NT group under both RB and DB, within-group comparisons are surprisingly similar. For example, at the FM level, T participants exhibit 0.83 ± 0.05 vs. 0.84 ± 0.06 (RB vs. DB), and NT participants show 0.66 ± 0.03 vs. 0.61 ± 0.03 . These results contrast with prior studies reporting a clear increase in CSF velocity during deep or forced breathing compared to natural/regular breathing.

=> Thank you for this important discussion. A difference in mean CSF speed was observed between the T and NT groups, whereas no difference was found between breathing conditions within each group. This likely reflects that CSF mean speed primarily represents cardiac-driven CSF motion. Notably, Liu et al. reported in real-time PC-MRI studies that during deep breathing compared with free breathing (their terminology corresponding to regular breathing), cardiac-driven CSF flow did not differ

significantly, while respiration-driven CSF flow increased markedly (by as much as 326%), reflecting a strong enhancement of the respiratory contribution to overall CSF dynamics ³.

Similarly, Yildiz et al. analyzed the respiratory CSF (rCSF) component using a bandpass filter of <0.6 Hz and the cardiac components (c1CSF and c2CSF) at 0.6–1.6 Hz and 1.6–2.7 Hz, respectively. They observed that deep abdominal breathing (DAB) compared with spontaneous breathing (SponB) increased rCSF but did not affect c1CSF or c2CSF ⁸.

Consistent with these findings, Dreha-Kulaczewski et al. applied a T₁-weighted gradient-echo–based real-time imaging approach that detects signal-intensity changes reflecting CSF motion (rather than quantitative velocity maps) and reported a pronounced forced inspiration effect when cardiac signals were filtered out and the data were averaged across the respiratory cycle ⁹.

Together, these studies support our interpretation that CSF mean speed primarily reflects the cardiac component, whereas displacement and net flow may capture the respiratory modulation effects. However, this still does not fully explain why the T group exhibited higher baseline CSF mean speed. One possible explanation is that long-term breathing practice may influence autonomic or vascular tone ¹⁰, as also suggested by our finding of greater respiratory sinus arrhythmia (HR displacement) in the trained group during RB, resulting in subtle changes in cardiac pulsatility that may secondarily affect CSF movement. Future studies incorporating detailed assessments of cardiorespiratory function in the trained group, such as HRV analyses, will be needed to clarify the physiological basis of these group differences, and this point has been incorporated into the revised Discussion.

“A difference in CSF mean speed was observed between the T and NT groups, whereas no difference was found between breathing conditions within each group. This likely reflects that CSF mean speed primarily represents cardiac-driven CSF movements. Liu et al. reported that during deep breathing compared with free breathing, cardiac-driven CSF flow did not differ significantly, while respiration-driven CSF flow increased markedly (by up to 326%), indicating an enhanced respiratory contribution to overall CSF dynamics ³. Similarly, Yildiz et al. showed that deep abdominal breathing increased respiratory CSF component but not cardiac components ⁸, and Dreha-Kulaczewski et al. observed a strong inspiration effect when cardiac signals were removed ⁹. Together, these findings support that CSF mean speed largely reflects the cardiac component, while displacement and net flow capture respiratory modulation. The higher baseline CSF mean speed in the trained group may relate to long-term breathing training effects on autonomic or vascular tone ¹⁰, as supported by our finding of greater HR displacement (RSA) during RB. Future studies including HRV and cerebrovascular reactivity are needed to clarify this mechanism ¹¹.”

Looking at a representative participant in Figure S12, the flow rate amplitude appears similar between RB and DB, although the phase is clearly different. Moreover, the DB condition seems more reflective of slow breathing rather than deep breathing, which may explain the absence of increased amplitudes. Or, perhaps these results from the use of mean speed compared to max/min or peak-to-peak CSF velocity, which as indicated previously raises a major concern for the methods, thus findings.

=> Thank you, and we agree with the reviewer's observation. The cardiac pulsation-driven amplitude, represented by CSF mean speed, did not differ markedly between RB and DB, as this metric primarily reflects cardiac-related CSF velocity oscillations. In contrast, the slower, respiration-driven bulk flow component, captured by CSF displacement, showed clear differences between RB and DB, consistent with enhanced respiratory modulation during DB.

To better illustrate these distinctions, we have updated Figure S14 and S15 to include one representative participant from the NT group and one from the T group, both selected from the top 50% of the FM net flow distribution. The revised figure now displays the CSF velocity time series, the absolute CSF velocity time series, CSF flow rate time series, and CSF volume displacement time series, providing a clearer visualization of both cardiac and respiratory contributions under each breathing condition.

6. "In this study participants in the T group performed the 'breathing practice' component of these sessions."

7. Does it mean only the T participants performed the deep breathing based on training style, but the NT group performed a different deep breathing pattern? Or both groups performed same instructions? Please clarify.

=> We thank the reviewer for requesting clarification. Both groups were instructed identically. During RB, participants followed their natural respiratory pattern, and for the DB, they were asked to breathe slowly and deeply while avoiding hyperventilation or breath-holding. The main instruction for both conditions was to engage in a breathing pattern that felt natural and sustainable. During DB, trained participants likely performed this more efficiently due to their prior experience. The main philosophy of Seokmun Hoheup breathing practice is to promote natural, lower belly-centered breathing, which can be described as a slow and deep breathing. With continued practice, this breathing naturally develops into fine, long, and deep respiration with an inhale-to-exhale ratio of approximately 6 to 4.

This effect seems to be represented in the group differences in chest and diaphragm displacement shown in Figure S13. During RB, T participants showed greater diaphragm displacement compared with NT participants, while chest displacement did not differ between groups. Lung area displacement was also higher in the T group in

RB. During DB, T participants exhibited minimal chest wall movement with greater diaphragm excursion, whereas NT participants relied on both chest wall motion and diaphragm contribution. However, the diaphragm displacement during DB did not reach statistical significance between groups ($p = 0.3073$), and there was no difference in lung area displacement.

These findings provide insight into the training effect, with T participants primarily engaging the diaphragm as the main driver of respiration, whereas NT participants exhibited both thoracic and diaphragm breathing pattern.

8. Please provide phantom study section prior to human subject research methodology for the validation of the technique, and readability of the study flow.

=> Done. Thank you.

9. Fig S3d Condition 6: Is the overlapping phase signature (unlike in Conditions 4-5)- due to reversed signals during two measurements?

=> Thank you for the clarification. Multiple scan time series were previously overlapped in the original figure. In the revised Figure S2, we now display only one representative scan time series to ensure clarity.

10. Please provide Figures and Supplementary figures in the order mentioned in the manuscript.

=> Thank you. The supplementary figures have been updated to match the main figures and follow the overall flow of the manuscript.

11. Flow analysis (for the CSF ROI selection and extraction of CSF velocity time series): The manuscript currently lacks clarity regarding the process for selecting the CSF ROI and the CSF signal components, as well as the bandwidths applied. Specifically:
- It is stated that a bandwidth of [0.8–2 Hz] was used primarily for cardiac-driven CSF ROI selection, and [0.01–0.5 Hz] for the respiratory band. However, this does not explain what bandwidth was ultimately used for computation of the CSF velocity time series, and the derived measure of mean CSF speed, and derived cardiac- and respiratory CSF components. This raises a major methodological concern about whether different voxels were used for cardiac- vs. respiratory-driven CSF signals. If so, please justify why.

=> Thank you for this important comment. We recognize that the Methods section was not sufficiently clear on these points, and we have now updated it to clarify the ROI selection process. Voxels were selected using voxel-wise FFT analysis to identify those exhibiting cardiac-frequency oscillations (0.8–2.0 Hz). However, no bandpass filtering was applied to the actual PC-MRI velocity data after ROI selection. All subsequent analyses, including computation of CSF velocity time series and CSF mean speed, were performed using the full, unfiltered velocity signal.

Please clarify explicitly:

1. Was CSF signal ROI selection different for cardiac and respiratory components? If yes, why?

=> Thank you. No. The same CSF ROI was used to derive all CSF features.

2. What bandwidth was used for the computation of CSF velocity – to then derive mean CSF speed, which is reported as one primary outcome.

=> Thank you. No bandwidth filtering was applied. CSF velocity and CSF mean speed were calculated from the full, unfiltered velocity time series.

3. What bandwidth was used for the cardiac component of CSF signals, and what bandwidth was used for the respiratory component? Not just for voxel selection but for the actual analysis?

=> Thank you. The respiratory-driven effects were calculated using the respiratory belt signal to segment the CSF data into individual respiratory cycles³.

- The methodological choice of using CSF speed rather than CSF velocity also requires clarification as indicated before. RT-PCMRI provides velocity which preserves directionality, whereas speed removes this directional information and reduces interpretability. In phantom flow experiments, velocity is reported, but in the human experiments the manuscript uses the term speed. Please clarify this discrepancy.

=> Thank you for requesting clarification. The flow phantom data were analyzed using the same method, and the terminology has been updated to “mean speed” for consistency across both phantom and human experiments.

- Finally, since the study investigates two distinct breathing types (RB and DB) and diaphragm movement, it would be more appropriate to quantify maximum CSF (systolic), minimum CSF (diastolic), and/or peak-to-peak CSF velocities rather than mean speed. Averaging CSF speed obscures and likely underestimates the true magnitude of the instantaneous maxima and minima (as in Fig S14a CSF velocity time series and red line showing mean CSF speed) during RB vs DB, which then likely underestimates the true impact of breathing on CSF dynamics during two breathing patterns. Since this study relies on comparing RB & DB among T & NT groups, it is critical to compare actual dynamic range of CSF.

=> We thank the reviewer for this thoughtful comment. In real-time PC-MRI, each acquisition contains multiple cardiac cycles, and peak detection or maximum–minimum extraction across cycles would still require averaging to obtain a representative measure. This approach ultimately provides results comparable to the CSF mean speed. When respiratory belt data are used to segment the respiratory cycle, the maximum and minimum values within each cycle reflect the slower bulk flow component, represented by CSF displacement and net flow. Therefore, we believe that the combination of CSF mean speed, CSF displacement, and CSF net flow effectively

captures both the cardiac-driven and respiration-driven components of CSF dynamics across breathing conditions.

- Lastly, this methodological choice may explain why mean CSF speed appears very similar between RB and DB conditions, as indicated in one of my comments above-. And, I suspect/ suggest the reported “mean CSF speed” may reflect the cardiac-driven component of CSF motion rather than instantaneous CSF, which does not differ substantially between breathing patterns in this study.

=> We agree with the reviewer’s observation. The lack of within-group differences in CSF mean speed between RB and DB likely reflects that CSF mean speed primarily represents the cardiac-driven component of CSF motion, which remains relatively stable across breathing conditions. This interpretation is consistent with previous reports showing that CSF velocity is less affected by respiratory influences^{3,8,9}. In our new Supplementary Figure S5, we conduct a sensitivity analysis controlling for respiratory rate. When respiratory rate was regressed out, other CSF features showed reduced significance, whereas CSF mean speed and mean peak velocity remained largely unchanged. This further supports that CSF mean speed primarily reflects the cardiac-driven component of CSF motion rather than respiration-related effects.

12. Request for time- and frequency-domain representation of instantaneous CSF velocities: To aid methodology and interpretability, I strongly recommend that the authors provide representative examples of instantaneous CSF velocity time series and corresponding frequency domain signals. Specifically, please include:

-Time- and frequency-domain plots of band-pass filtered CSF velocities within [0–2 Hz] if that’s what used for CSF speed computations, for one participant from each group/condition (T-RB, T-DB, NT-RB, NT-DB).

- Time-domain velocity traces should be presented as in Fig. S14, with the instantaneous CSF velocity time series and the mean CSF speed indicated by a red line.

-The corresponding frequency-domain plots should be shown alongside, highlighting the dominant respiratory and cardiac peaks.

A similar figure is already included (Fig. S12), but it shows flow rates rather than velocity time series. Providing the velocity-based plots will substantially improve clarity, please demonstrate how filtering was applied, and allow readers to better understand how the reported outcomes were derived, which will remove the methodological concerns regarding the interpretability and discussion of findings.

=> Thank you for this helpful suggestion. We have added CSF velocity time series for both T and NT participants in Figure S14 and S15 and created a new supplementary figure showing the corresponding frequency-domain data. These additions provide representative time- and frequency-domain examples that clarify how the reported outcomes were derived. As bandpass analysis was not used in our main results, the bandpass-filtered data were removed from the sample case. Instead, we focused on the velocity time series and their corresponding frequency-domain representations derived

directly from the unfiltered velocity data.

13. An expert opinion on diaphragm motion analysis may be needed.

=> Thank you. Our analysis focused on the posterior region of the diaphragm, where the greatest excursion typically occurs during respiration ¹². Because the imaging plane was positioned slightly off-center near the right lung apex, the diaphragmatic muscle anchored around the L1–L3 vertebral levels exhibited pronounced downward motion during inspiration, reflecting posterior diaphragmatic movement. To accurately quantify this motion, we marked the posterior diaphragmatic contour along the dome and the diaphragm sulci connecting to the chest and back walls. The detailed description has been added to the Methods section.

“Diaphragm displacement was computed using the same approach. Among the three diaphragm landmarks, diaphragm 3, located in the posterior region, was used, as this region has been reported to exhibit the greatest excursion during respiration ¹². Because our imaging plane was positioned slightly off-center near the lung apex, the diaphragmatic muscle anchored around the L1–L3 vertebral levels exhibited prominent downward motion during inspiration, reflecting posterior diaphragmatic movement. To accurately quantify this effect, we marked the posterior diaphragmatic contour along the dome and the diaphragm sulci connecting to the chest and back walls.”

14. Please clarify the selection/inclusion of CSF metrics for correlation.

=> Thank you for requesting clarification. The CSF metrics used for correlation analysis were derived from the same set of features obtained during the flow analysis, including CSF mean speed, mean peak velocity, mean valley velocity, maximum and minimum flow rate, displacement, and net flow. These features were selected to represent both velocity-based and volume-based characteristics of CSF dynamics. The relationships among all features are shown in Supplementary Figure S10 and summarized in Supplementary Table S2.

15. T group practitioners (including authors included) who are aware of the study aims may intentionally or unintentionally exaggerate or modify their breathing during scanning, potentially amplifying apparent group differences.

=> We thank the reviewer for this important comment. All participants received identical instructions and were equally informed of the study aims. The initial participants who are current authors were not aware of the data results and participated under the same standardized protocol as other participants. However, given that Seokmun Hoheup-trained individuals have long-term breathing experience, intentional or unintentional modulation of breathing during scanning cannot be entirely excluded.

CSF movement is an involuntary physiological process, whereas respiration can be voluntarily modified in multiple ways, including breathing length, frequency, depth, volume, and the inhale–exhale ratio. Because it was not known which specific

respiratory features would most strongly influence CSF movement, all participants received standardized instructions: “During regular breathing, participants followed their natural respiratory pattern. For the deep breathing phase, they were asked to breathe slowly and deeply while avoiding hyperventilation or breath-holding.”

Interestingly, group differences were observed even during RB, suggesting that trained participants naturally maintain slower and deeper breathing patterns at rest. This was not part of our original hypothesis but provides an intriguing direction for future investigation. We have added this discussion to the revised manuscript and noted that future double-blind studies will be necessary to further rule out this potential bias.

16. And more broadly, as with many MRI-based physiological studies, it is not entirely clear whether breathing patterns in the MRI match the practitioners true practice patterns. These should be acknowledged in the Discussion, ideally within a defined Limitations section, for transparency.

=> Thank you for this valuable comment. We acknowledge that breathing patterns during MRI may not fully reflect the practitioners’ true practice patterns due to the scanning environment and physical constraints. In addition, during MRI scanning, respiratory signals were recorded using a pressure-balloon belt and fixated strap positioned on the upper abdomen just below the sternum. This setup may have limited full chest and abdominal expansion, potentially altering the natural breathing depth or pattern. This limitation has been noted in the Discussion.

“Another possible limitation of this study is that trained participants, who have long-term experience with controlled breathing, may have intentionally or unintentionally modulated their respiration during scanning, which could have influenced group differences. Although all participants received identical instructions and followed the same standardized protocol, minor differences in breathing behavior cannot be entirely ruled out. In addition, breathing patterns during MRI may not fully reflect the practitioners’ natural breathing patterns due to the confined scanning environment and physical constraints. Respiratory signals were recorded using a pressure-balloon belt with a fixation strap positioned on the upper abdomen just below the sternum, which may have restricted full chest and abdominal expansion and slightly altered breathing depth or rhythm. These factors should be considered when interpreting the findings, and future studies using blinded designs and less restrictive setups may help better isolate true training-related effects.”

Reviewer #2 (Remarks to the Author):

The authors adequately addressed my comments. Congratulations!

Reviewer #3 (Remarks to the Author):

Overview

The manuscript now presents a more refined study integrating phantom validation, physiological recordings, and MRI analysis to examine how breathing practices influence CSF dynamics. The Results are more clearly structured, with improved consistency and better integration of supplementary figures into the main text. The definitions of CSF flow features and the use of SEM with correlation analyses strengthen the mechanistic interpretation, and the phantom validation remains a key strength. The revisions address earlier concerns around data reuse and organization, which improves readability and focus. Some questions remain, particularly regarding how robustness of the SEM is reported and how multiple comparisons are handled, but the clarifications made so far represent clear progress.

=> We appreciate the reviewer's assessment and note the recognition that the integration of phantom validation, physiological recordings, and MRI analysis strengthens the study. We are glad the restructuring of Results and supplementary figures improved clarity and focus, and that the refinements to CSF feature definitions and SEM analyses were seen as useful. We will address the remaining points regarding SEM robustness and multiple comparisons in detail below.

Major comments

1. In group comparisons (Fig. 3, S11), trained participants show longer inhale length, greater diaphragm displacement, and increased HR and SSS displacement during regular breathing. A potential concern is that the significantly lower respiratory rate in the trained group could partly account for these differences. The authors note that both HR and respiratory rate were incorporated into the SEM and multivariate models as mechanistic features rather than uncontrolled confounders, which is a reasonable approach. To strengthen confidence in this interpretation, it may be helpful to make this distinction clearer in the text and, if possible, to include a simpler sensitivity analysis (for example, adjusting group comparisons for respiratory rate) to demonstrate that the findings are robust across different analytical frameworks.

=> Thank you for this thoughtful comment. We performed a sensitivity analysis regressing out respiratory rate. The results show that CSF mean peak velocity and CSF mean speed was not affected by removing the respiratory rate effect. At the lateral ventricle (LV), group differences in displacement and net flow were no longer significant, particularly during regular breathing. At the foramen magnum (FM), the regular breathing difference disappeared, while the deep breathing difference between groups remained. These findings are consistent with our model results showing that LV dynamics are mainly driven by respiration, whereas FM reflects both respiratory and additional autonomic influences.

In this study, participants were instructed to follow their natural respiratory pattern and maintain a rhythm that felt comfortable and sustainable, rather than attempting to breathe as slowly or as long as possible. It is important to note that untrained attempts to slow breathing excessively can elevate carbon dioxide (CO₂) levels, activating central chemoreceptors in the brainstem and inducing irregular or unstable breathing patterns¹³. We have added these results as a new supplementary figure and expanded the Discussion accordingly.

Fig. 1 CSF flow group differences between breathing practice-trained (T) and non-trained (NT) participants.

Figure S6. Figure S6. Sensitivity analysis controlling for respiratory rate (RR).

Fig 2. Modeling the path from respiration to CSF net flow.

“Results

A sensitivity analysis was performed by regressing out RR (Fig. S6). CSF mean peak velocity and CSF mean speed was not affected by this adjustment. At the FM, the RB difference disappeared, while the DB difference between groups remained. At the LV, group differences in displacement and net flow were no longer significant, particularly during RB.

Discussion

These findings align with the model results, indicating that LV dynamics are primarily driven by respiration, whereas FM reflects both respiratory and additional autonomic influences. The analysis highlights the importance of respiratory rate but also shows that FM differences during deep breathing persist, suggesting involvement of mechanisms beyond simple respiratory modulation. Additionally, the trained group exhibited a slower respiratory rate even during regular breathing, supporting a training-related adaptation rather than an intentional effort to breathe slowly. Participants were instructed to maintain a natural and sustainable breathing rhythm without consciously controlling their rate. This distinction is important, as untrained attempts to reduce respiratory rate can elevate CO₂ levels and induce irregular or unstable respiration¹³. The observed effects therefore reflect a trained, physiologically integrated breathing pattern, rather than simply slower breathing.”

2. The tertile-based analysis (Fig. 4, S13) is a creative way of moving beyond training status and looking for generalisable features, but tertiles may not be the most stable approach given the modest sample size. It might be worth clarifying whether alternative splits (e.g., quartiles or median) produce comparable results.

=> Thank you for this helpful suggestion. Our previous “tertile” grouping represented a split at the mean between the upper and lower 50% of participants. After adopting the reviewer’s suggestion, we re-tested using the median, which produced an identical division. As this more accurately reflects our analysis, we have updated the Results and Methods sections to indicate that the grouping was based on the median.

More broadly, the statistical handling of multiple comparisons remains a little unclear. The study reports many tests (t-tests, paired comparisons, regressions, correlations, and Kruskal–Wallis post hoc analyses) but no correction method (such as false discovery rate or Bonferroni) is described. Without adjustment, the risk of false positives is increased significantly.

=>We appreciate the reviewer’s concern regarding multiple comparisons. In this study, our analyses were designed as a sequence of sub-analyses progressing from broader physiological characterization to more specific CSF measures, following the principle of a closed testing procedure. At each stage, we observed highly significant group differences, many of which remain significant even under a conservative Bonferroni adjustment (e.g., uncorrected $p < 0.0001$), which supports proceeding to the next level of analysis.

To address the area with the greatest risk of inflated false positives, we applied Bonferroni correction across the full correlation matrix (15 features, 225 pairwise tests; Supplementary Fig. S10). For the other comparisons (e.g., group and within-subject contrasts, regression models), we evaluated results within their hierarchical context. Once omnibus significance was established, individual comparisons were interpreted directly, similar in logic to Fisher's LSD framework. This balances statistical rigor with interpretability while avoiding unnecessary loss of power.

In addition, although the Methods indicate that SEM model fit was assessed with indices like RMSEA, CFI, TLI, and SRMR, these values are not reported in the Results. Including the fit indices would make it much easier for readers to judge the adequacy of the causal framework.

=> We appreciate this suggestion. Our analysis was carried out using a two-step modeling approach combining multivariate regression and SEM to examine directional relationships among respiratory, physiological, and CSF flow features. This hierarchical regression-based design provides clarity on the role of individual features while reducing interpretational confounding in the joint model. Because this framework is regression-based, it does not yield conventional global SEM fit indices (CFI, TLI, RMSEA, SRMR), which are defined only for a single joint covariance-based SEM. To make this clear, we have revised the Methods to describe the modeling framework explicitly and, in the Results, we report the standardized coefficients and R^2 for each regression that composes the diagram.

In addition, to incorporate the reviewer's suggestion, we have included exploratory joint SEMs in the Supplement for transparency. For the FM model, the joint SEM yielded $\chi^2(19)=104.4$, CFI=0.87, TLI=0.69, RMSEA=0.22, SRMR=0.16. For the LV model, the joint SEM yielded $\chi^2(19)=75.7$, CFI=0.90, TLI=0.76, RMSEA=0.19, SRMR=0.10. These indices reflect both the restrictive nature of the specified path framework and collinearity among predictors, but the path estimates were consistent with those obtained from our hypothesis-driven hierarchical regression framework.

3. The observation of a small but positive mean net CSF flow at the foramen magnum (~0.05 ml; Figs. S13–S14) is intriguing but raises interpretive challenges. Taken at face value, this could suggest net upward CSF transport from the spinal canal, which would have important implications given ongoing debate about CSF production sites. At the same time, the Methods indicate that factors such as ROI placement, baseline drift, and integration of oscillatory signals could introduce subtle offsets, and while phantom validation shows good reproducibility in controlled conditions, it cannot fully rule out small in vivo artifacts. For this reason, it would be valuable for the manuscript to discuss this finding explicitly, considering both the possible physiological interpretation and the methodological caveats, and to situate it within the broader literature so that readers do not over-interpret the result.

=> We appreciate the reviewer's thoughtful comment regarding the small but positive mean net CSF flow observed at the foramen magnum. In response, we have revised the Discussion to address both the possible physiological interpretation and the

methodological caveats. The revised text notes that 2D PC-MRI measures velocity at a single imaging plane and may not fully represent bulk CSF circulation, and that while phantom experiments demonstrated good reproducibility, small *in vivo* offsets may still arise from technical factors such as low-frequency phase drift, gradient calibration shifts, or ROI placement variability.

“Accurately measuring net CSF flow remains challenging. Most prior studies have emphasized oscillatory peak velocities rather than net displacement, and although net flow has been reported, few investigations provide comprehensive measurements over extended durations^{14,15}. Importantly, net flow derived from 2D PC-MRI may not fully represent the true bulk circulation, as the technique measures velocity across a single imaging plane rather than whole-brain transport. To address potential technical offsets inherent to MRI acquisition, including low-frequency phase drift and gradient calibration shifts, we applied detrending to the CSF data prior to secondary analysis. Our phantom experiments further demonstrated good reproducibility under controlled conditions. To minimize variability in ROI placement, we implemented a semi-automatic procedure to make region identification more objective. In our analysis, detrending and segmentation by respiratory cycles further constrain the metric to reflect only the cardiorespiratory-driven component of CSF movement, rather than long-term circulation.”

Minor comments

1. Figure S11 highlights group differences in chest (thoracic) versus diaphragm displacement during deep breathing, but these are only briefly mentioned in the text. A short explanation linking these patterns to different breathing strategies would help clarify their relevance.

=> Thank you. We have expanded the text to clarify the relevance of the group differences in chest and diaphragm displacement shown in Figure S13. During regular breathing (RB), trained participants showed greater diaphragm displacement compared with non-trained participants, while chest displacement did not differ between groups. Lung volume displacement was also higher in the trained group. During deep breathing (DB), trained participants exhibited minimal chest wall movement with greater diaphragm excursion, whereas non-trained participants relied more on chest wall motion with less diaphragm contribution. However, the diaphragm displacement during DB did not reach statistical significance between groups ($p = 0.3073$), and there was no difference in lung volume displacement. These findings provide insight into the training effect, with trained participants primarily engaging the diaphragm as the main driver of respiration, whereas non-trained participants exhibited a more thoracic breathing pattern. This explanation has been added in the result section of the revised manuscript.

“During RB, T participants showed greater diaphragm displacement compared with NT participants, while chest displacement did not differ between groups. Lung volume displacement was also higher in the T group. During DB, trained participants exhibited minimal chest wall movement with greater diaphragm excursion, whereas NT

participants showed greater chest motion and less diaphragm contribution. The diaphragm displacement during DB did not reach statistical significance between groups ($p = 0.3073$), and no difference was observed in lung volume displacement. These results indicate that T participants primarily relied on diaphragm motion, whereas NT participants exhibited a more thoracic breathing pattern (Fig. S13)."

2. The representative case in Figure S12 includes only a trained participant. Here, adding a contrasting non-trained example would make the claim of altered coupling with practice more compelling.

=> Thank you. We have added CSF velocity traces for both trained and non-trained participants in Figure S14 and S15.

3. Several supplementary figures, particularly S11 and S13, contain results central to interpretation yet are only briefly referenced. The authors should weave these figures more directly into the main text which would reduce the need for readers to flip back and forth.

=> Thank you for this helpful suggestion. We aimed to keep the main figures focused on the core group comparisons to maintain clarity and readability. The chest displacement data from Supplementary Figure S13 have been moved to the main figure to better integrate key findings. The remaining supplementary figures contain complementary but overlapping data and were retained to avoid overcomplicating the main figure.

4. The phantom validation would benefit from separating cardiac-like and respiratory-like oscillations into distinct subsections for clarity.

=> Thank you for the helpful suggestion. We have updated the phantom validation section for better clarity. Figure S2a–c has been merged into Figure S17, which now presents separate columns for the cardiac-like and respiratory-like oscillations.

5. The supplementary videos appear to quantify respiratory area using straight-line approximations, which cut-out the natural curvature of the breathing trace. While the Methods make clear that quantitative diaphragm motion analysis was performed with DeepLabCut, it remains ambiguous whether the simplified videos were used quantitatively or merely for illustration. This distinction should be clarified, as linear approximations could underestimate displacement.

=> We thank the reviewer for this excellent point. The change in respiratory area was quantified by measuring the area difference between inhalation and exhalation (Figure S20d). We agree that a linear estimation of diaphragm motion can underestimate displacement, as the diaphragm is an asymmetric dome that flattens during inspiration and becomes more curved during exhalation. A linear approximation therefore captures only partial motion.

The greatest excursion occurs at the dome of the diaphragm, making analysis of this distance essential. However, the dome's motion must be evaluated relative to the diaphragm sulci, which remain largely stationary as they are tethered to the chest wall.

Since the absolute position of the sulci can shift slightly with chest wall motion, the analysis was designed to measure dome excursion relative to sulci movement.

6. The link to glymphatic function in the Discussion is interesting, though since glymphatic clearance (along perivascular spaces) was not directly measured, it may help to phrase this connection a bit more cautiously. Clarifying that the present findings reflect macroscopic CSF dynamics, which may provide context for but are distinct from perivascular clearance processes, would strengthen the interpretation without overextending the scope.

=> Thank you. We agree with the reviewer's comment and have revised the Discussion to clarify this point. The connection to glymphatic function is now described more cautiously, emphasizing that the present findings reflect macroscopic CSF dynamics measured with MRI, which are distinct from perivascular clearance processes. We note that while our results may provide physiological context for understanding glymphatic function, they do not directly assess glymphatic or perivascular clearance. This clarification has been added to the revised manuscript.

"As current clinical strategies to modulate CSF flow are primarily invasive, and there is emerging interest in applying noninvasive approaches¹⁶, these results position respiratory training as a feasible and promising intervention to support CSF and glymphatic clearance, especially in individuals with compromised cardiovascular function. Future studies will be needed to directly connect respiration-driven CSF net flow with glymphatic function, and dementia-focused investigations are planned to determine whether these effects translate into clinical benefit."

To "As current clinical strategies to modulate CSF flow are primarily invasive, and there is emerging interest in applying noninvasive approaches¹⁶, these results position respiratory training as a feasible and promising intervention to support macroscopic CSF dynamics, especially in individuals with compromised cardiovascular function. Future studies will be needed to clarify how respiration-driven CSF movements relates to glymphatic function, and dementia-focused investigations are planned to determine whether these effects translate into clinical benefit."

Minor errors

1. A few errors should be corrected to improve accuracy and readability. For example, in the section on baseline physiology, the heart rate for deep breathing in the trained group is reported as "66.0 ± 0.16 bpm." "LV CS displacement" is written instead of "LV CSF displacement." The formatting of CSF-related acronyms varies across sections.

=> Thank you. We have corrected these errors and double-checked all CSF-related acronyms for consistency throughout the manuscript.

References

- 1 Karki, P. *et al.* Real-Time 2D Phase-Contrast MRI to Assess Cardiac- and Respiratory-Driven CSF Movement in Normal Pressure Hydrocephalus. *J Neuroimaging* **35**, e70000 (2025). <https://doi.org/10.1111/jon.70000>

- 2 Spijkerman, J. M. *et al.* Phase contrast MRI measurements of net cerebrospinal fluid flow through the cerebral aqueduct are confounded by respiration. *J Magn Reson Imaging* **49**, 433-444 (2019). <https://doi.org/10.1002/jmri.26181>
- 3 Liu, P. *et al.* Cardiac and respiratory activities induce temporal changes in cerebral blood volume, balanced by a mirror CSF volume displacement in the spinal canal. *Neuroimage* **305**, 120988 (2025). <https://doi.org/10.1016/j.neuroimage.2024.120988>
- 4 Dreha-Kulaczewski, S. *et al.* Respiration and the watershed of spinal CSF flow in humans. *Sci Rep* **8**, 5594 (2018). <https://doi.org/10.1038/s41598-018-23908-z>
- 5 Liu, P. *Acquisition and processing of phase contrast Magnetic Resonance Imaging for the quantification of cerebral blood and cerebrospinal fluid flow under respiratory influence Acquisition et traitement de l'imagerie par Résonance Magnétique en contraste de phase pour la quantification des écoulements cérébraux du sang et du Liquide Cérébro-Spinal sous influence respiratoire*, Université de Picardie Jules Verne, (2021).
- 6 Takizawa, K., Matsumae, M., Sunohara, S., Yatsushiro, S. & Kuroda, K. Characterization of cardiac- and respiratory-driven cerebrospinal fluid motion based on asynchronous phase-contrast magnetic resonance imaging in volunteers. *Fluids Barriers CNS* **14**, 25 (2017). <https://doi.org/10.1186/s12987-017-0074-1>
- 7 Wang, Y. *et al.* Cerebrovascular activity is a major factor in the cerebrospinal fluid flow dynamics. *Neuroimage* **258**, 119362 (2022). <https://doi.org/10.1016/j.neuroimage.2022.119362>
- 8 Yildiz, S. *et al.* Immediate impact of yogic breathing on pulsatile cerebrospinal fluid dynamics. *Sci Rep* **12**, 10894 (2022). <https://doi.org/10.1038/s41598-022-15034-8>
- 9 Dreha-Kulaczewski, S. *et al.* Inspiration is the major regulator of human CSF flow. *J Neurosci* **35**, 2485-2491 (2015). <https://doi.org/10.1523/JNEUROSCI.3246-14.2015>
- 10 Zaccaro, A. *et al.* How Breath-Control Can Change Your Life: A Systematic Review on Psycho-Physiological Correlates of Slow Breathing. *Front Hum Neurosci* **12**, 353 (2018). <https://doi.org/10.3389/fnhum.2018.00353>
- 11 Liu, P. *et al.* Synchronous assessment of CSF and cerebral arteriovenous flow interactions across ultra-low, low, respiratory, and cardiac frequencies using real-time phase-contrast MRI. *NeuroImage* **321**, 121490 (2025). <https://doi.org/https://doi.org/10.1016/j.neuroimage.2025.121490>
- 12 Aktas, G. *et al.* Spinal CSF flow in response to forced thoracic and abdominal respiration. *Fluids Barriers CNS* **16**, 10 (2019). <https://doi.org/10.1186/s12987-019-0130-0>
- 13 Russo, M. A., Santarelli, D. M. & O'Rourke, D. The physiological effects of slow breathing in the healthy human. *Breathe (Sheff)* **13**, 298-309 (2017). <https://doi.org/10.1183/20734735.009817>
- 14 Dreha-Kulaczewski, S. *et al.* Identification of the Upward Movement of Human CSF In Vivo and its Relation to the Brain Venous System. *J Neurosci* **37**, 2395-2402 (2017). <https://doi.org/10.1523/JNEUROSCI.2754-16.2017>

- 15 Lindstrom, E. K., Ringstad, G., Mardal, K. A. & Eide, P. K. Cerebrospinal fluid volumetric net flow rate and direction in idiopathic normal pressure hydrocephalus. *Neuroimage Clin* **20**, 731-741 (2018). <https://doi.org/10.1016/j.nicl.2018.09.006>
- 16 Jin, H. *et al.* Increased CSF drainage by non-invasive manipulation of cervical lymphatics. *Nature* **643**, 755-767 (2025). <https://doi.org/10.1038/s41586-025-09052-5>

Reviewer #1

Remarks to the Author:

The authors have adequately addressed the previous review comments. The revisions improved the clarity and quality of the manuscript. I have no additional comments at this time unless other reviewers have, and support moving forward with publication. Congratulation to the entire study team.

Reviewer #3

Remarks to the Author:

The manuscript is much improved and the comments have been mostly addressed.

We had one remaining point in regards to the SEM model fit as part of the original major comment 2:

Including the exploratory joint SEMs and their fit indices is appreciated and adds transparency. That said, the reported values (e.g., CFI below 0.9, RMSEA above 0.1) suggest that the overall model fit was not very strong. While the explanation about collinearity and model restrictions is reasonable, it would be good to acknowledge this more directly so that readers understand the limitation.

A simple sentence noting that the joint SEMs showed poorer global fit, and that this limits the strength of causal interpretation even though the directional trends were consistent, would address this clearly.

=> We appreciate the reviewer's thoughtful feedback and careful attention to the model fit issue. We agree that the joint SEMs showed a suboptimal global fit in the exploratory joint SEM (CFI < 0.90; RMSEA > 0.10), limiting the strength of causal interpretation even though the directional trends were consistent. We have now acknowledged this limitation in the Discussion section of the revised manuscript.

"Second, the modeling showed a suboptimal global fit in the exploratory joint SEM (CFI < 0.90; RMSEA > 0.10), limiting the strength of causal interpretation even though the directional trends were consistent."